# Above- and belowground biodiversity jointly tighten the P cycle in agricultural grasslands

Yvonne Oelmann [1✉], Markus Lange [2], Sophia Leimer [3], Christiane Roscher[4,5], Felipe Aburto [6], Fabian Alt[1], Nina Bange[1], Doreen Berner[7], Steffen Boch [8], Runa S. Boeddinghaus [7], François Buscot [9], Sigrid Dassen[10], Gerlinde De Deyn [10,11], Nico Eisenhauer [5,12], Gerd Gleixner[2], Kezia Goldmann [9], Norbert Hölzel[13], Malte Jochum [5,12], Ellen Kandeler[7], Valentin H. Klaus [14], Till Kleinebecker [15], Gaëtane Le Provost [16], Peter Manning [16], Sven Marhan[7], Daniel Prati[17], Deborah Schäfer[17], Ingo Schöning[2], Marion Schrumpf[2], Elisabeth Schurig[1], Cameron Wagg [18,19], Tesfaye Wubet [5,20] & Wolfgang Wilcke [3]

Experiments showed that biodiversity increases grassland productivity and nutrient exploitation, potentially reducing fertiliser needs. Enhancing biodiversity could improve P-use efficiency of grasslands, which is beneficial given that rock-derived P fertilisers are expected to become scarce in the future. Here, we show in a biodiversity experiment that more diverse plant communities were able to exploit P resources more completely than less diverse ones. In the agricultural grasslands that we studied, management effects either overruled or modified the driving role of plant diversity observed in the biodiversity experiment. Nevertheless, we show that greater above- (plants) and belowground (mycorrhizal fungi) biodiversity contributed to tightening the P cycle in agricultural grasslands, as reduced management intensity and the associated increased biodiversity fostered the exploitation of P resources. Our results demonstrate that promoting a high above- and belowground biodiversity has ecological (biodiversity protection) and economical (fertiliser savings) benefits. Such win-win situations for farmers and biodiversity are crucial to convince farmers of the benefits of biodiversity and thus counteract global biodiversity loss.

[1] Geoecology, University of Tübingen, Tübingen, Germany. [2] Max Planck Institute for Biogeochemistry, Jena, Germany. [3] Institute of Geography and Geoecology, Karlsruhe Institute of Technology (KIT), Karlsruhe, Germany. [4] UFZ - Helmholtz Centre for Environmental Research, Physiological Diversity, Leipzig, Germany. [5] German Centre for Integrative Biodiversity Research (iDiv) Halle-Jena-Leipzig, Leipzig, Germany. [6] LISAB, Dep. Silvicultura, Universidad de Concepción, Concepción, Chile. [7] Institute of Soil Science and Land Evaluation, Soil Biology Department, University of Hohenheim, Stuttgart, Germany. [8] WSL Swiss Federal Research Institute, Birmensdorf, Switzerland. [9] UFZ - Helmholtz Centre for Environmental Research, Soil Ecology Department, Halle, Germany. [10] Department of Terrestrial Ecology, Netherlands Institute of Ecology, Wageningen, The Netherlands. [11] Department of Environmental Sciences, Soil Biology, University of Wageningen, Wageningen, The Netherlands. [12] Leipzig University, Institute of Biology, Leipzig, Germany. [13] Institute of Landscape Ecology, University of Münster, Münster, Germany. [14] Institute of Agricultural Sciences, ETH Zürich, Zürich, Switzerland. [15] Institute of Landscape Ecology and Resource Management, Justus-Liebig-University Gießen, Gießen, Germany. [16] Senckenberg Biodiversity and Climate Research Centre (SBiK-F), Frankfurt, Germany. [17] Institute of Plant Sciences, University of Bern, Bern, Switzerland. [18] Department of Evolutionary Ecology and Environmental Studies, University of Zürich, Zürich, Switzerland. [19] Fredericton Research and Development Center, Agriculture and Agri-Food Canada, Fredericton, NB, Canada. [20] UFZ - Helmholtz Centre for Environmental Research, Community Ecology Department, Halle, Germany. ✉email: yvonne.oelmann@uni-tuebingen.de

Experiments have demonstrated that high biodiversity can increase ecosystem functioning and service provision[1–3]. Increased productivity of plant species mixtures, in comparison to their component species in monocultures, is generally attributed to facilitation and niche complementarity of species associated with a more complete exploitation of resources[4]. Nutrients such as nitrogen (N) and phosphorus (P) are exploited more exhaustively with increasing diversity of plant mixtures[5–9]. Consequently, less N is leached to the groundwater[10,11]. Therefore, high biodiversity contributes to a tight N cycle (contrary to a broken cycle associated with element loss[12]). However, P leaching was not reduced by increased plant species diversity[8], partly because of the lower water solubility of mineral species of P compared to N, which forced plants to develop alternative P acquisition strategies beyond the direct uptake from soil solution by roots[13]. The low P concentrations in soil solution challenges the study of the P cycle. Accordingly, we lack a mechanistic understanding of biodiversity effects on the P cycle despite the importance of P as a limiting nutrient in terrestrial ecosystems in response to high N deposition[14] and decreasing mineral P fertiliser availability, which has been identified as a threat to global food security[15].

In contrast to N, P resource acquisition by nearly all plant species in the temperate climate zone is linked to a symbiosis with mycorrhizal fungi (arbuscular mycorrhizal fungi [AMF] in grasslands and ectomycorrhizal fungi [EMF] in forests)[16]. The plant-AMF symbiosis involves the direct transfer of P from the fungus to the plant. Species identity and the richness of plant and mycorrhizal fungal species influence productivity[16,17] and thus, potentially the extent of P transfer. Furthermore, P acquisition of plants can also originate from an indirect transfer via the microbial community such as the release of P during turnover of the microbial P pool[18] or from enzymatically catalyzed release from soil organic matter[19]. Since all three processes (P acquisition by plants via fungi, microbial and enzyme-mediated P release) were previously shown to increase with increasing plant species richness if studied separately[8,19,20], we hypothesise that effects of biodiversity—including plant and mycorrhizal species richness—on the P cycle are mediated by plant-microbe interactions.

While the mechanisms that underlie biodiversity effects on the P cycle in controlled grassland experiments are partially understood, it remains unknown whether the same mechanisms operate in permanent, managed agroecosystems such as agricultural grasslands. Approaching this knowledge gap from an experimental perspective, biodiversity and management can be manipulated simultaneously. Biodiversity experiments comprising management measures showed that positive biodiversity effects on productivity persisted on top of management effects[21–23]. Furthermore, in such crossed biodiversity-fertilisation trials the exploitation of N resources by the plant community increased with increasing diversity of plant mixtures irrespective of fertilisation[24,25]. However, already during the short-term duration of these experiments, the authors mentioned that the seeded species composition (and richness) changed depending on the management treatments[23,24]. In long-established agricultural grasslands, biodiversity and management effects interact even more strongly[26]. Management drives biodiversity because it can be considered as an environmental filter of the community composition[27]. Grassland management directly or indirectly filters plant species composition because fertilisation, mowing, and grazing change abiotic site conditions. Accordingly, plant and AMF species richness are reduced by high management intensity associated with high resource availability in soil[28–30]. At the same time, the few plant species growing under high resource availability are known for their large productivity and thus, exploitative nutrient acquisition[28,31]. Whether resource exploitation in agricultural grasslands is dominated by biodiversity effects, i.e. the more diverse the more exhaustive is the resource exploitation, or by direct management effects, i.e. the more intensive and the more productive the more exhaustive is the resource exploitation, awaits to be shown. However, biodiversity and management are closely intertwingled, with high nutrient concentrations resulting in lower plant biodiversity in the long run, so that a fully crossed experiment cannot be maintained for very long in reality. We therefore suggest that the combination of insights gained from both, biodiversity experiments and observational data of agricultural grasslands represents an alternative promising avenue.

Here, we made use of the opportunity to combine results from 76 grassland mixtures of a biodiversity experiment with those from 100 permanent agricultural grasslands along a management gradient. All grasslands were located on soils developed from calcareous bedrock, which can be considered representative of large parts of Central Europe. We were particularly interested in the extent to which P resources are exploited by organisms (P exploitation, i.e. the part of the total bioavailable P pool in the ecosystem that is stored in the biomass) and whether P exploitation is related to biodiversity. We related organismic P stocks (in the plant and microbial community) to the total bioavailable P pool in the system (sum of both organismic P stocks and bioavailable P stocks in the soil [$HPO_4^{2-}$ and $H_2PO_4^-$ and organically bound P in labile and moderately labile P fractions accessible with moderate chemical extractants including $NaHCO_3$ and dilute NaOH])[8]. We first tested mechanisms underlying the biodiversity-P cycle relationship in the biodiversity experiment. Second, we explored whether biodiversity effects on the P cycle and the underlying mechanisms detected in the biodiversity experiment persisted along a management gradient (in terms of fertilisation, mowing, and grazing intensity). To do so, we used structural equation modelling (SEM) to set up an a priori defined mechanistic model of biodiversity effects on P exploitation in the biodiversity experiment and applied this model to data from the agricultural grasslands.

In this work, we show that in agricultural grasslands, management effects either overrule or modify the driving role of plant biodiversity observed in the biodiversity experiment. Nevertheless, greater above- (plants) and belowground (mycorrhizal fungi) biodiversity contributes to tightening the P cycle in agricultural grasslands. Our results demonstrate that promoting a high above- and belowground biodiversity has ecological (biodiversity protection) and economical (fertiliser savings) benefits.

## Results and discussion

**Explanation of the P exploitation**. Based on the knowledge that has been gathered so far for biodiversity effects on single processes of the P cycle (Table 1), we set up an SEM describing the relationships between biodiversity and P exploitation for the biodiversity experiment (Supplementary Fig. 1). We considered variables that are known to influence the extent to which P resources can be exploited including (i) the soil organic carbon ($C_{org}$) stocks linked to microbial biomass[32,33] and (ii) AMF species richness and relative abundance governing the symbiotic transfer of P from microbes to plants[16,17]. Because of the close correlation between AMF species richness and relative abundance, we introduced these two variables with correlated errors in the SEM.

We found that P exploitation was positively related to plant species richness and this relationship was mediated by $C_{org}$ stocks and microbial P stocks in soil (Fig. 1A). Under diverse plant mixtures, an increase of $C_{org}$ stocks in soil likely resulted from an increased C input from above- and belowground plant litter and

**Table 1 Expected relationships underlying the structural equation models of the biodiversity experiment and the agricultural grasslands.**

| Regression | Expected slope | Reason | Reference |
|---|---|---|---|
| Plant species richness – AMF species richness | ↑ | Host specificity, AMF functional trait complementarity, mycelial networks | van der Heijden et al.[17]; Hiesalu et al.[45]; Dassen et al.[73] |
| AMF species richness – Microbial P stocks | ↑ | More complete resource use achieved by diverse AMF community; inferred from AMF abundance – microbial P stocks relationship (see below) because abundance and species richness were closely related (Supplementary Table 1) | van der Heijden et al.[79] |
| AMF species richness – Plant P stocks | ↑ | More complete resource use achieved by diverse AMF community and P subsequently provided to plant host | van der Heijden et al.[17]; Frew[80] |
| Plant species richness – AMF abundance | ↑ | More niches for mycorrhizal colonization in diverse plant communities | van der Heijden et al.[17]; Hedlund et al.[81] |
| AMF abundance – Microbial P stocks | ↑ | More complete resource use achieved by more abundant AMF community | van der Heijden et al.[79] |
| AMF abundance – Plant P stocks | ↑ | More complete resource use achieved by more abundant AMF community and P subsequently provided to plant host | van der Heijden et al.[17]; Treseder et al.[82]; Köhl et al.[83] |
| Plant species richness – $C_{org}$ stocks | ↑ | Increased above- and belowground plant litter input | Fornara & Tilman[84]; Steinbeiss et al.[85]; Lange et al.[34] |
| $C_{org}$ stocks – Microbial P stocks | ↑ | $C_{org}$ serves as an energy source and microbial homoeostasis requires a concomitant uptake of P | Hacker et al.[86]; Sorkau et al.[32] |
| $C_{org}$ stocks – Plant P stocks | ↑ | $C_{org}$ stocks in soil are positively linked to plant biomass which forms part of plant P stock calculations | Fornara & Tilman[84]; Steinbeiss et al.[85]; Lange et al.[34] |
| Microbial P stocks – P exploitation | ↑ | Increased P exploitation if microbes take up and store more P | |
| Plant P stocks – P exploitation | ↑ | Increased P exploitation if plants take up and store more P | Oelmann et al.[81] |
| Management intensity – Plant species richness | ↓ | Environmental filtering: under high resource availability, dominant outcompete other species | Allan et al.[28] |
| Management intensity – P exploitation | ↑ | Environmental filtering selects exploitative species | Allan et al.[28] |

↑ = positive slope of regression, ↓ = negative slope of regression, $C_{org}$ = organic carbon in soil.

root exudates[34]. Given that $C_{org}$ serves as the energy source for the heterotrophic soil microbial community, $C_{org}$ concentrations were reported to be related to microbial C concentrations[34,35]. Consequently, plant species richness was also shown to be positively correlated with microbial C concentrations[34,35]. In turn, microbial C stocks were related to microbial P stocks, because microorganisms typically take up P proportional to their biomass to maintain nutrient homoeostasis[36]. Our results may suggest nutrient homoeostasis of the microbial community since the ratios of microbial C and P (Cmic:Pmic) were not significantly related to plant species richness (Supplementary Table 1). Accordingly, our model indicates that a positive effect of plant species richness on P exploitation was related to the link between the P and the C cycle that allowed soil microorganisms to exploit P resources in soil more exhaustively.

Similarly, plant P stocks increased with increasing plant species richness (Fig. 1A). Plant P concentrations were not related to plant species richness while plant biomass was (Supplementary Table 1). In line with our mere bivariate correlation, increased biomass production of diverse grassland mixtures is well documented[37,38]. Biomass production is linked to the C cycle and thus, the simultaneous plant species richness effects on biomass production and plant P stocks reflect the coupling of plant species richness effects on the C and the P cycle. However, the biomass-induced increased P demand of the more diverse plant communities was apparently not met by the AMF symbiosis. Although plant species richness and AMF species richness and relative abundance were related (Fig. 1A), this relationship did not directly feed back to the P cycle. There was neither a significant path connecting AMF species richness or relative abundance and plant P stocks (Fig. 1A) nor a significant relationship between plant P concentrations or plant biomass and AMF species richness or relative abundance (Supplementary Table 1). Instead, P release during microbial turnover might replenish the P pool in soil that is available for uptake by plants[18]. Our results from a recent study in the same biodiversity experiment corroborate the role of microbial turnover for the available P pool in soil[20].

In conclusion, our data suggest that plant species richness influenced the C cycle both above- and belowground which then translated into plant species richness effects on the P cycle. Ultimately, the more diverse the plant community was, the more P resources were exploited by plants and by soil microorganisms in the biodiversity experiment.

**Interactions of biodiversity and management.** In a first approach, we applied the SEM describing biodiversity effects on the P cycle in the biodiversity experiment to the observational data collected in the agricultural grasslands with the additional

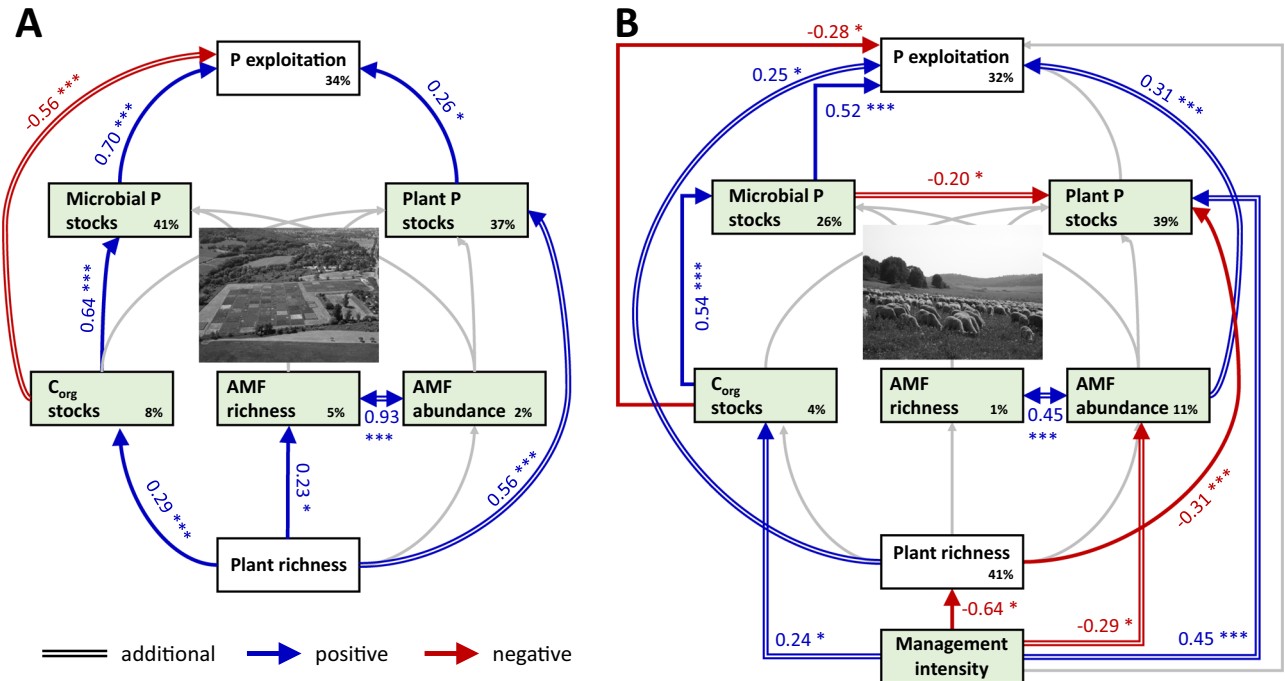

**Fig. 1 Role of biodiversity for the P cycle.** Panel (**A**) refers to the biodiversity experiment and Panel (**B**) shows the agricultural grasslands. Biodiversity comprises plant and AMF (arbuscular mycorrhizal fungi) species richness. In the structural equation model (SEM), blue and red arrows display relationships with positive and negative slopes, respectively. Grey arrows indicate non-significant relationships. Solid lines of arrows refer to relationships expected according to the hypotheses (Table 1), whereas parallel lines of arrows indicate paths that were included in addition (selection procedure described in Methods). Round-shaped paths refer to the (driving) role of biodiversity while square paths indicate the role of management. Numbers on arrows indicate standardised path coefficients. Percentages in boxes give the explained variance. *$p < 0.05$, **$p < 0.01$, ***$p < 0.001$. The SEMs matched with the data: (**A**) Fisher's C = 7.12, $p = 0.93$, $df = 14$; (**B**) Fisher's C = 12.17, $p = 0.43$, $df = 12$. $C_{org}$ = organic carbon, P = phosphorus. Photo credits: The Jena Experiment (**A**), Jörg Hailer (**B**).

consideration of management intensity (Supplementary Fig. 2). This model contained management intensity as an influential variable for plant species richness which then exerted the same influence as in the biodiversity experiment model. In addition, we inserted a direct relationship between management intensity and P exploitation. However, this initial SEM did not adequately represent the data from the agricultural grasslands (Supplementary Fig. 2).

We therefore included additional paths to optimise the model (see Methods and additional paths in Fig. 1). In the optimised model for the agricultural grasslands, four out of seven variables were directly related to management (Fig. 1B). The direct and positive influence of management intensity on plant P stocks was likely caused by P from fertilisers[39]. Furthermore, management intensity contributed to the explanation of $C_{org}$ stocks of agricultural grasslands (Fig. 1B). On the one hand, high management intensity commonly goes along with high livestock numbers and heavy machinery use and thus, increases soil bulk density[40]. Because soil bulk density is used to calculate stocks (see Methods), increased soil bulk density translates into increased stocks. On the other hand, high management intensity can be associated with the application of organic fertilisers e.g., manure and slurry containing $C_{org}$ and nutrients[41]. Compared to the biodiversity experiment, the effect of plant species richness was replaced by the influence of management on $C_{org}$ stocks in the agricultural grasslands. In line with the considerations on the link between $C_{org}$ stocks and microbial metabolism outlined above for the biodiversity experiment, a greater management intensity was also associated with increased microbial P stocks and thus, with an enlarged P exploitation (Fig. 1B). Similar to plants[28], P-exploitative microbial species seem to gain dominance under management-induced

increased P availability[42]. In general, future studies should focus on the relation of microbial taxa and their functioning in the P cycle[43] which could encompass the study of functional genes.

Low management intensity was related to high plant species richness (Fig. 1B). In addition, low management intensity was associated with low plant P stocks (Fig. 1B) via a decrease in plant biomass (Supplementary Table 2) because of decreased fertilizer input. These relationships are in line with well-documented beneficial effects of reduced management intensity on plant species richness[29]. In addition, high plant species richness was linked to low plant P stocks (Fig. 1B) which likely is a result of a higher portion of plant species with low biomass production and low nutrient concentrations in these plant communities because of a conservative resource use[28]. Accordingly, low management intensity was linked to plant P stocks directly via management measures and indirectly via filtering effects in agricultural grasslands that specifically remove plant species with a conservative resource use if land-use intensity increases. As a consequence, the relationship between plant species richness and plant P stocks in agricultural grasslands (Fig. 1B; 14 to 56 plant species per grassland plot) was reversed as compared to the positive, biomass-driven coefficient in the biodiversity experiment (Fig. 1A) covering species richness levels from monoculture to complete native communities (1-, 2-, 4-, 8-, 16- and 60-species mixtures). Furthermore, the significant relationship between plant P stocks and P exploitation in the biodiversity experiment disappeared in the agricultural grasslands (Fig. 1). Instead, there was a negative relationship between plant P stocks and microbial P stocks (Fig. 1). We suspect that microbes partly outcompeted plants in terms of P uptake[44] which might have been reinforced by less cooperative AMF species that preferentially retain P for

themselves[30,45] and thus, increased microbial P stocks at the expense of plant P stocks. Yet, the positive relationship between plant species richness and P exploitation observed in the biodiversity experiment stayed the same in the agricultural grasslands: Increasing plant species richness increased P exploitation (Fig. 1). However, the mechanism underlying the plant species richness effect on P exploitation in the biodiversity experiment (i.e., the coupling of the C and P cycles above- and belowground) did not apply to the agricultural grasslands. Instead, the direct link between plant species richness and P exploitation implies that variables representing the underlying mechanism were not included in our model of the agricultural grasslands.

The less intense the management was, the more abundant and diverse were the AMF communities in the agricultural grasslands (Fig. 1B). Decreased P availability in soil due to reduced fertilisation is known to increase the species richness and abundance of the fungal community and in particular of the AMF community[29,30]. We were able to show that this increase in AMF abundance was associated with an increased P exploitation (Fig. 1B). The management effect on total P exploitation was mainly mediated via AMF and plant species richness (combined effect of all paths leading to P exploitation via AMF and plant species richness = −0.22), while the combined effect via $C_{org}$ stocks was negligible (0.002). Therefore, management effects in agricultural grasslands either overruled and modified the role of plant species richness observed in biodiversity experiments. Nonetheless, reduced management intensity was beneficial for above- and belowground biodiversity and for the exploitation of P resources in soil. Here we showed that both, above- and belowground biodiversity contribute to tightening the P cycle in agricultural grasslands.

**Biodiversity theory put into practice**. From an applied perspective, even small positive effects of biodiversity on productivity and P exploitation could yield benefits in terms of savings of fertilisation including labour and energy costs for application of the fertiliser. Such economic arguments should help to convince farmers to promote and conserve biodiversity. The balance between benefits of reduced management for biodiversity and potentially detrimental effects on yields remains a challenging issue and requires open-minded solutions. The identification of win-win situations is crucial to merge opposing views of representatives of agriculture – arguably one of the leading causes of biodiversity loss[46] – and nature conservation. Only if these two too frequent opponents act in concert, biodiversity theory will be put into practice and could help counteract global biodiversity loss and develop sustainable management strategies.

## Methods

**Study sites**. Grassland biodiversity experiment: Our data originate from the Jena Experiment (http://www.the-jena-experiment.de)[47]. The field site is located near the German city of Jena (50°55′ N, 11°35′ E; 130 m above sea level). Mean annual air temperature is 9.9 °C, and mean annual precipitation amounts to 610 mm[48]. The soil is an Eutric Fluvisol[49] developed from up to 2 m-thick fluvial sediments that are almost free of stones. Sediments largely originated from the same geological series (Pleistocene loess on limestone [Anisian/Ladinian] of the Middle Triassic) as described for one of the agricultural grassland regions (Hainich-Dün, see below). The systematic variation in soil texture as a consequence of fluvial dynamics was considered in the experimental design by arranging the experimental plots in four blocks at different distances to the river. The study site was converted from grassland to arable land in the early 1960s and used for cropping until the establishment of the experiment in 2002.

The main experiment comprised 82 plots that were split in different subplots with a core area of approximately 43.5 m². Each plot contains a specific combination of plant species (1, 2, 4, 8, 16, 60 species) that belong to different numbers (1, 2, 3, 4) of plant functional groups (grasses, [non-leguminous] small herbs, [non-leguminous] tall herbs, legumes). The species were chosen from a pool of 60 species typically found in mesic Molinio-Arrhenatheretea meadows. Each

plant species richness level had 16 replicates except for 14 mixtures with 16 species and 4 replicates of the 60-species mixture. Monocultures of *Bellis perennis* L. and *Cynosurus cristatus* L. included in the original design had to be abandoned because of poor establishment. We excluded the 60-species mixtures from statistical analyses (final $n = 76$), because they contain the complete species pool and thus, would potentially strengthen the sampling effect. To maintain the sown species richness levels, plots were weeded three times a year by carefully removing weeds and their roots. This minimises potential bias from weed root contributions. The management of all plots was adapted to extensive meadows used for hay production i.e., mown but not fertilised. All plots were mown twice a year in June and September with the harvest being removed from the plots.

Agricultural grasslands: We studied long-established, agriculturally managed (i.e. not experimentally assembled) grasslands in an interdisciplinary, large-scale and long-term programme, the Biodiversity Exploratories (http://www.biodiversity-exploratories.de)[50]. The Biodiversity Exploratories comprise three study regions in Germany (Schwäbische Alb, Hainich-Dün, and Schorfheide-Chorin). We aimed to disentangle biodiversity and management effects and therefore, had to remove additional confounding factors such as geologic parent material (calcareous versus non-calcareous). Because of substantial differences in the geologic parent material[50] and of P fractions in soil[51], we excluded the Schorfheide-Chorin from our analyses. Standardised field plots were located in the Schwäbische Alb and in the Hainich-Dün, both middle mountain ranges in Germany. The Schwäbische Alb is located between 460–860 m above sea level, has an annual mean temperature of 6–7 °C and an annual precipitation of 700–1,000 mm[50]. The geologic parent material is calcareous bedrock of the Oxfordian Age (Epoch: Upper Jurassic) from which Leptosols and Cambisols have developed[49]. The Hainich-Dün is located between 258–550 m above sea level, has an annual mean temperature of 6.5–8 °C and an annual precipitation of 500–800 mm[50]. The geologic parent material in the Hainich-Dün is calcareous bedrock of the Middle Triassic Epoch partly covered by Pleistocene loess from which Cambisols, Stagnosols, Vertisols and Luvisols have developed[49]. The design of the Biodiversity Exploratories was described in detail by Fischer et al.[50] We used 100 grassland plots (50 in each of Schwäbische Alb and Hainich-Dün) which can be classified as meadows (mown one to four times per year but not grazed), pastures (grazed but not mown), and mown pastures (both mown and grazed). Plots were selected to represent a gradient of management intensity with different fertilisation, frequency of mowing, and livestock units.

**Management intensity**. Management intensity of the biodiversity experiment has been the same since the establishment (no fertilisation, no grazing, mown twice a year). In the agricultural grasslands, management intensity (number of livestock units per hectare per year, frequency of mowing events per year, and amount of N-fertiliser applied per hectare per year) varied among plots and was assessed yearly via questionnaires answered by the farmers[52]. Notably, the plots were not amended with mineral P fertiliser except for two plots which received 25 kg ha⁻¹ P in 2011. However, manure and slurry were applied and also contain labile P fractions which might transform into sparingly soluble P minerals in the long run[53]. To account for the variation in management of the agricultural grasslands, we used data of the sampling campaigns in 2011 and 2014 as described below. We first measured and calculated variables for each year individually. There were robust correlations between years for the different variables (correlation coefficient of up to $r = 0.86$) and therefore, we used means of the two years for further evaluation. We considered the mean land-use intensity of the years 2011 and 2014 as more robust than the single measurements alone, because the P availability in soil is determined by the site conditions and the long-term land use, particularly in the absence of mineral P fertilization as was the case at all our study sites, except two.

**Plant species richness and biomass harvest**. In the biodiversity experiment, sown species richness was determined on a 9 m² (3 m × 3 m) area in May 2014. In agricultural grasslands, vascular plant species richness was assessed on a 16 m² (4 m × 4 m) area by sampling all species from mid May to mid June in 2011 and 2014. In order to account for the difference in survey area between experimental and agricultural grasslands, a separate survey of 9 m² subplots nested within the 16 m² subplots was conducted in April and May 2018 in a selected number of plots in the agricultural grasslands ($n = 18$). Species richness differed significantly between 9 m² and 16 m² subplots (mean number of species ± standard error; 9 m²: 22 ± 2; 16 m²: 24 ± 2; paired t-test: T = −5.15; p < 0.001). Accordingly, we used a scaling factor of 0.91 i.e., the slope of the regression of species richness on 9 m² on species richness on 16 m² (Supplementary Fig. 3), to adjust species richness of all agricultural grasslands to the area of species richness measurements in the biodiversity experiment.

In the biodiversity experiment, aboveground plant biomass of target species was harvested in May 2014, at the estimated peak biomass before mowing. Aboveground biomass was sampled in all plots within a frame (0.2 × 0.5 m, height 0.03 m) at two random locations per plot and sorted by target species, weeds and detached dead plant material. Biomass was extrapolated to 1 m². In agricultural grasslands, biomass of a 2 m² large area was harvested as mixed samples of eight quadrats of 0.25 m² in close proximity to the area in which the vegetation was recorded from mid May to mid June in 2011 and 2014. Temporary fences ensured that biomass was sampled at peak standing crop but without any effect of possible

mowing or grazing events. Detached dead material was excluded from biomass sampling. Biomass was assessed on a dry-weight basis (drying at 70–80 °C for 48 h) in the experimental and agricultural grasslands.

**Soil sampling.** Soil in the biodiversity experiment was sampled in April 2014 for analyses of $C_{org}$ concentrations and bulk density. In each plot, three soil cores were taken to a depth of 0.3 m using a split-tube sampler (4.8 cm diameter). Soil cores were segmented into 5 cm-depth sections and pooled per depth sections and plot[34]. We used the mean $C_{org}$ concentrations of the upper three intervals (0 to 0.15 m). For the measurement of P in soil of the biodiversity experiment, sampling took place in September 2013. Nine soil cores per plot with a diameter of 0.02 m were taken at a depth of 0 to 0.15 m and combined to a composite sample considered representative for the plot. Although these samples were not taken during the identical period of time as for the agricultural grasslands (see below), we tested whether the results are applicable to the growing season of 2014. First, bioavailable Pi concentrations in soil were closely correlated between years (September 2013 and October 2014; $r = 0.88$, $p < 0.001$, $n = 79$). Second, resin-extractable P concentrations on samples of the 2013 campaign measured in the laboratory matched with P released under field conditions in May 2014 ($r = 0.40$, $p < 0.001$, $n = 78$).

In agricultural grasslands, soil samples were collected simultaneously within two weeks in May 2011 and May 2014. Samples were taken along two orthogonal transects of 20 m. Sampling points were shifted by 0.5 m in 2014 compared with 2011 to avoid an overlap of sampling positions. In each plot, 14 samples from 0 to 0.1 m soil depth were taken using core augers (diameter ~52 mm). Samples were mixed, cooled and transported to a field lab where they were sieved (<2 mm), all within 8 h of sampling. All measurements described below for agricultural grasslands were done on aliquots of samples of these joint sampling campaigns.

**Laboratory analyses and calculations.** Air-dry plant material sampled in the experimental and agricultural grasslands was ground with a mill using a 0.5-mm screen for chemical analyses. Nitrogen concentration of plant material was measured in ground samples with an elemental analyser for the biodiversity experiment and near-infrared spectroscopy (NIRS) for the agricultural grasslands. Plant samples were digested in a microwave with concentrated nitric acid and hydrogen peroxide[54] and P concentrations were determined by inductively-coupled plasma optical emission spectrometry (ICP-OES). In 2011, P concentrations were measured by NIRS in biomass of the agricultural grasslands. We multiplied the latter data by 0.9844 to match with the digestion method based on data of the year 2009 for which both the digestion and NIRS method had been applied ($P_{digestion} = 0.9844 \times P_{NIRS}$, $r = 0.80$, $p < 0.001$, $n = 98$). Plant samples collected in 2014 in the agricultural grasslands were analysed for P concentrations by means of an X-ray fluorescence spectrometer. We can exclude a methodological shift in plant P concentrations between years, because digestion followed by ICP-OES analyses and XRF analyses were shown to match well[55]. Nitrogen:P ratios in plant material were calculated on an elemental mass basis. We calculated P stocks in aboveground biomass by multiplying biomass [g m$^{-2}$] harvested in May with their P concentrations [mg g$^{-1}$] and with the number of mowing events. In this way, we likely overestimated the absolute values of aboveground plant P stocks. But at the same time, we ensured comparability between experimental and agricultural grasslands because, for the latter, plant material of mowing events later than May was not available.

In the experimental and agricultural grasslands, soil pH values were determined with a glass electrode in a 1:2.5 soil:0.01 M CaCl$_2$ water suspension. Total C concentrations were determined on ground air-dry samples by an elemental analyser. Organic C concentrations were calculated by subtracting inorganic C concentrations (determined after removal of $C_{org}$ at 450 °C in a muffle furnace) from total C concentration. Soil bulk density was calculated by weighing a 100-cm³ core after drying the soil at 40 °C.

In the experimental and agricultural grasslands, various P fractions in soil were measured according to the method of Hedley et al.[56] modified by Kuo[57]. The sequential extraction scheme had three steps (bioavailable P (NaHCO$_3$ extractable), moderately labile P (NaOH extractable), and mineral P (HCl extractable)). The bioavailable P fraction comprises H$_2$PO$_4^-$/HPO$_4^{2-}$ ions in soil solution and those weakly adsorbed to mineral surfaces. More strongly adsorbed P to iron and aluminium oxides and (oxy)hydroxides is recovered in the moderately labile fraction. The mineral P fraction contains P bound in apatite and other Ca phosphates[56,58,59]. In all extraction solutions, Pi concentrations were analysed using the ammonium molybdate-ascorbic acid blue method described by Murphy and Riley[60] and measured with a continuous flow analyser. Total dissolved P concentrations in NaHCO$_3$- and NaOH-extracts were measured with an ICP-OES. For the labile and moderately labile fractions (NaHCO$_3$-P, NaOH-P), organic P concentrations were calculated by subtracting Pi from total dissolved P concentrations in the extracts.

In the experimental and agricultural grasslands, microbial P was measured by hexanol fumigation using a combination of the methods of McLaughlin et al.[61] and Kouno et al.[62]. Three subsamples of each soil sample were prepared by adding deionised water and one anion-exchange membrane. One of the three subsamples was additionally mixed with hexanol as fumigation reagent ($P_{Hex}$) and one with a P spike ($P_{Spike}$), while nothing was added to the last subsample ($P_{H_2O}$). Nitric acid was used to exchange the P adsorbed onto the membranes. Phosphate concentrations in solutions were measured with a continuous flow

analyser. We calculated microbial P concentrations as the difference between hexanol-fumigated and non-fumigated samples and accounted for P retention during extraction by including the P spike[63]. Because the calculated microbial P concentrations underestimate the amount of P stored in microbial biomass because of fumigation/extraction efficiency constraints e.g., with respect to gram-positive bacteria, we divided calculated microbial P concentrations by 0.4[62,64]. We calculated stocks of P fractions and microbial P in soil based on bulk density [g m$^{-2}$ (0.15 m soil depth)$^{-1}$] and multiplication with the respective P concentrations [mg g$^{-1}$]. Phosphorus exploitation was calculated as the contribution [%] of organismic P stocks (either in the aboveground part of plants or microbes) to the sum of bioavailable P stocks (both organismic P stocks + labile P stocks + moderately labile P stocks) according to Oelmann et al.[8]. The aboveground plant P stocks represent the plant demand of P that is removed with the harvest(s) each year. It is reasonable to assume that microbial P stocks in soil can also be regarded as the annual microbial P demand since it has been shown that the microbial P stock in soil turns over once every growing season irrespective of management[39].

In the biodiversity experiment, phosphomonoesterase (PAse) activity was measured according to the modified assay of Eivazi and Tabatabai[65]. For each soil sample, one replicate and one blank value were included in the laboratory study. We incubated soil samples with *p*-nitrophenylphosphate as an organic substrate for enzyme activity (pH = 11). *p*-nitrophenylphosphate was added to blanks only after incubation. Directly after filtration, *p*-nitrophenol concentrations were measured with a spectrophotometer. In agricultural grasslands, PAse activity was determined by fluorescence measures in a buffered solution of pH 6.1 after Marx et al.[66], as described in Berner et al.[67].

In the biodiversity experiment, the fungal-to-bacterial ratio was assessed by applying the phospholipid fatty acids (PLFA) method[68]. Within 48 h after sampling, the soil was kept at 4 °C, sieved to 2 mm, remains of roots were manually removed and the samples were stored at −20 °C until further sample processing. PLFA were extracted according to the method of Bligh and Dyer[69] as modified by Kramer and Gleixner[70]. As an indicator for fungal PLFA 18:2ω6,9 was used[71,72]. The bacterial PLFA was calculated as sum of the PLFA markers 14:0i, 15:0i, 15:0a, 16:0i, c16:1ω7c, 17:1, 17:0i, 17:0a and 18:1ω7 (Frostegard & Baath 1996). Furthermore, AMF species richness and relative abundance in the biodiversity experiment was investigated in soil samples (0–15 cm depth) collected in 2010 and analysed using DNA extraction and amplicon sequencing by 454-pyrosenquencing as described in Dassen et al.[73]. We amplified 18 S rRNA gene fragments from fungi and protists with primer FR 1 and the modified version of FF390 designed to also include the Glomeromycota comprising AMF[74] (Supplementary Table 3). In agricultural grasslands, AMF were identified based on DNA extraction. DNA was extracted from soil of each plot using the MO BIO Power Soil DNA isolation kit (MO BIO Laboratories, Carlsbad, CA, USA) following the manufacturer's protocol. Afterwards we used a nested PCR approach to amplify fungal 18S-rDNA by using the primer pairs GlomerWT0/Glomer1536[75] and NS31/AML2[76,77] (Supplementary Table 3), containing the Illumina adapter sequences. PCR products were then purified, cleaned and sequenced using Illumina MiSeq. The AMF sequences were processed using a customized bioinformatic pipeline following MOTHUR SOP (using Version 1.39.5) as implemented in DeltaMP (https://github.com/lentendu/DeltaMP). The taxonomical assignment was done against the MaarjAM database (https://maarjam.botany.ut.ee/). AMF OTUs were merged according to VT assignment. OTUs assigned only to genus level were kept as OTUs. Only those AMF appearing on more than five plots were considered. AMF species richness was calculated as the number of species including OTUs. The relative abundance of AMF was calculated by relating the reads per species/OTU to the total sum of reads across all plots. AMF appearing on less than five plots had low relative abundances (<1%) and thus, were considered to play a negligible role. Accordingly, only those AMF appearing on more than five plots were included. AMF species richness was calculated as the number of species including OTUs. The relative abundance of AMF was calculated by relating the reads per species/OTU to the total sum of reads across all plots.

**Statistical analyses.** Normal distribution of the variables and the homoscedasticity of the models were visually inspected and if necessary, variables were log-transformed to meet the prerequisite for statistical analyses (Supplementary Tables 1, 2). Pearson correlations were calculated applying the corr.test function in the R library psych[66]. The Pearson correlations were corrected for multiple inference using the Benjamini & Hochberg correction. Using the R library piecewiseSEM[78], confirmatory path analyses were applied to test the causal relationships between plant diversity and P exploitation in the biodiversity experiment and in the agricultural grasslands and how land-use intensity impacts this relation in the latter. To identify the processes underlying the plant diversity and the land use intensity effects on P exploitation, we set up conceptual path models for each, the biodiversity experiment and the agricultural grasslands (Supplementary Figs. 1 and 2; Table 1). The models include AMF species richness, relative AMF abundance, $C_{org}$ stocks, plant P stocks and microbial P stocks as possible mediators. To account for the block design of the biodiversity experiment and the two regions considered in the agricultural grasslands, block was fitted as a random factor in the biodiversity experiment models and region as a random effect in the agricultural grassland models. Furthermore, we checked for spatial autocorrelation within each region. To this end, we related the residuals associated to each variable in the SEM

to the geographic coordinates of each plot. Since we found no indications of relationships or clustering of the residuals, we assumed independent data within the regions of the agricultural grasslands. In order to test whether the conceptual models adequately represent the measured data Fisher's C test statistic was used. To obtain adequate models the conceptual models were updated stepwise by including missing paths as indicated by the tests of directed separation[78]. Details of the outcome of the SEMs in addition to that provided in Fig. 1 can be found in Supplementary Tables 4, 5 and 6.

**Reporting summary**. Further information on research design is available in the Nature Research Reporting Summary linked to this article.

## Data availability

This work is based on data from several projects of the Biodiversity Exploratories programme (DFG Priority Program 1374) and The Jena Experiment (DFG FOR 456 & 1451). The data used for analyses are publicly available from the Biodiversity Exploratories Information System (https://doi.org/10.17616/R32P9Q) and the jexis database (https://jexis.idiv.de/), respectively, or will become publicly available after an embargo period of five years from the end of data assembly to give the owners and collectors of the data time to perform their analysis. The raw AMF Illumina sequences for the Biodiversity Exploratories have been deposited in the National Center for Biotechnology Information (NCBI) Sequence Read Archive (SRA) under BioProject accession number PRJNA706003. Microbial community data of the Jena Experiment have been archived in the Pangaea database (https://doi.org/10.1594/pangaea.874990). Any other relevant data are available from the corresponding author upon reasonable request.

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

## Acknowledgements

We thank all people who helped with the setup and management of the Jena Experiment (biodiversity experiment) and in particular the initiators, E.-D. Schulze, B. Schmid, and W.W. Weisser. Thanks also to all the helpers who assisted during the weeding campaigns. The Jena Experiment is funded by the Deutsche Forschungsgemeinschaft (DFG, FOR 456 & 1451) with additional support from the Swiss National Science Foundation (SNF), the Friedrich Schiller University Jena and the Max Planck Society. NE acknowledges support by the German Centre for Integrative Biodiversity Research Halle–Jena–Leipzig, funded by the German Research Foundation (FZT 118). M.L. gratefully acknowledges the support of the Zwillenberg-Tietz Foundation. We further thank A. Hemp, S. Gockel, M. Gorke, K. Lorenzen, K. Reichel-Jung, S. Renner, I. Steitz, F. Straub, M. Teuscher, J. Vogt, S. Weithmann, K. Wells and K. Wiesner (local management teams), C. Fischer, M. Gleisberg, J. Mangels and S. Pfeiffer (central office), B. König-Ries, J. Nieschulze, A. Ostrowski and M. Owonibi (central database management), M. Fischer, D. Hessenmöller, E. Linsenmair, J. Nieschulze, E-D. Schulze, and the late E. Kalko for their roles in setting up the Biodiversity Exploratories program (agricultural grasslands), which was funded by the Deutsche Forschungsgemeinschaft Priority Program 1374 Infrastructure-Biodiversity Exploratories. Fieldwork permits were given by the responsible state environmental offices of Baden-Württemberg, Thüringen, and Brandenburg (according to §72 BbgNatSchG). We are grateful to Barbara Schmitt, Stefan Blaser, Verena Busch, Helmut Hillebrand and Anne Ebeling for sharing their data.

## Author contributions

Y.O. and W.W. conceived the idea of this study. Y.O. and M.L. contributed equally to the data analysis and writing of the manuscript. C.R., F.A. (Alt), N.B., D.B., S.B., R.S.B., F.B., S.D., G.D.D., N.E., G.G., K.G., N.H., M.J., E.K., V.H.K., T.K., G.L.P., P.M., S.M., D.P., D.S., I.S., M.S., E.S., C.W., T.W., namely all authors but F.A. (Aburto) and S.L. contributed data. Y.O., S.L. and M.L. performed the analyses. Y.O. performed the literature search and wrote the first draft of the manuscript and all the authors contributed substantially to the revisions.

## Competing interests

The authors declare no competing interests.
