## [Peer Review File · Nature Communications]

Reviewer comments, first round –

Reviewer #1 (Remarks to the Author):

This study tries to disentangle the complex relationships among soil nutrient status, plant diversity, and P cycling. This is a very relevant topic, and as the authors highlight, a better mechanistic understanding of the drivers affecting the link between plant diversity and P availability may hold important consequences for the optimization of agricultural practices both in terms of biodiversity conservation and fertilizer reduction. I was therefore quite excited reading the title of the manuscript. However, after reading it, I was left with the impression that the data obtained and analyses performed, while commendable in many ways, did not justify the strength of the claim made by the title.

I see several issues with this study, some fundamental in nature and others more practical.

First, the only manipulated variable across all plots studied is plant diversity. Overall nutrient richness or P availability were never manipulated. This is not a criticism, as I very well understand the difficulty of manipulating multiple factors especially in a large-scale field setting, but an observation. It necessarily implies that any relationships linking plant diversity on the one hand to ultimate response variables on the other, are essentially correlative. There is nothing wrong with correlative work, and I believe SEM is the ideal tool to explore hypotheses about the modulating influences of intermediate variables. However, care should be taken in claiming causality based on such correlative work. A step in the right direction is applying a hypothetical model based on one set of data to a separate data set for validation, as the authors did. In this case, however, the model performed poorly on the new data. The authors' response was to build a new model including a factor (management) they thought could explain the discrepancy. This is a good next step, but unfortunately, the authors did not proceed by testing their new, now better fitting model including management on a separate validation data set as they did with their first model but seemed to consider this model "final". The authors nonetheless explain many of the patterns they see in their data by the effect of management, which was poorly validated and again, not a manipulated factor. In other words, the final model based on the agricultural plots is entirely correlative and fitted to the data rather than confirmatory. I therefore urge the authors to treat it as such. This means using much more care in drawing conclusions from these models, which are exploratory at best.

In addition to these rather fundamental comments, I have a few issues regarding the way in which data were analyzed. I refer to my detailed comments below but summarize the essence here. First, in the SEM of the agricultural grasslands, a case could be made for several logical pathways that were not included. If there were valid reasons for not including these, they should be given and explained. Second, I do not see why many of the conclusions are drawn from the table of rough pairwise Pearson correlations provided in Fig. S4., especially about variables mediating relationships among other variables, for which (another) SEM would make more sense. Third, for the experimental plots, which were likely randomized in space, this is less of an issue, but the plots in the agricultural fields are clearly not independent, as they are clustered (likely at multiple levels) in space. Violating the assumption of independent data will lead to erroneous conclusions regarding any analyses in which the dependence structure is not properly taken into account. To my understanding, this was done neither in the SEM neither in the Pearson correlations. Fourth, running a very large number of significance tests on Pearson correlations will result in about 5% of them turning out significant purely by chance. I know there is quite some discussion about the merits of multiple testing corrections, but in this case it seems advisable.

Finally, I found the writing often a tad overgeneralizing and vague. For further details I refer to my individual comments listed below.

Detailed comments:

- The abstract is confusing without reading the remainder of the manuscript, yet should be understandable as a self-contained piece of text. Roughly summarized, it states that a) plant biodiversity stimulates P cycling in nutrient-rich soils, b) P availability has a negative effect on plant biodiversity. This immediately begs the question what is meant by nutrient-rich. P is also a nutrient. If "nutrients" are good for the plant diversity - P cycle relationship but P itself is bad, does this mean "nutrients other than P"? Then the sentence "Our results demonstrate that promoting a high plant biodiversity on nutrient-rich soils and a high soil microbe, particularly AMF, biodiversity on nutrient-poor soils has ecological (biodiversity protection) and economical (fertiliser savings) benefits." It is not clear from the preceding sentences why promoting plant biodiversity has to happen on nutrient-rich soils and promoting AMF on nutrient-poor soils and not the other way around. I suggest you rephrase the abstract to ensure that each sentence logically builds on previous statements.
- Line 57. "profits" should be "profit" as a mass noun, or possibly "benefits"
- Line 68. I suggest changing "nutrient" to "nitrogen" here. Otherwise the "other nutrients" in the next sentence makes little sense.
- Line 70. I would think terrestrial ecologists have been aware of the critical role of P for quite a long time, so I am not sure whether there is still a "growing awareness".
- Line 76. I suggest changing "mycorrhizal species" to "mycorrhizal fungi species".
- Line 77-79: This sentence lacks some clarity. Does "symbiotic partner" still refer to AMF here, or to the wider soil microbial community? Please clarify.
- Line 92. "two way" should be "two-way"
- Line 96. Filter of what? Of plant species composition? Please clarify.
- Line 98. Not only small-scale, I would argue.
- Line 100. This sentence is ambiguous too, as "different directions" can refer to a difference among filters or a difference between effects on plants and effects on soil organisms. I assume the latter is meant here, but please rephrase for clarity.
- Line 102. The meaning of this sentence also evades me. How does the fact that mechanisms may have been overlooked follow from the fact that prior positive biodiversity effects on productivity have been reported? What is the logic here? It reads as if you imply that scientists only look for mechanisms of negative effects and not of positive ones.
- Line 128. I would advise the authors to be careful in their wording regarding SEMs. I suggest changing "originated from" to "related to". Also, the total path strength of plant species richness to bioavailable P mediated by organic C was 0.35. The total path via microbial P had a coefficient of only 0.14. The negative path via phosphatase activity in the soil had a coefficient of -0.15. The negative link between plant species richness and bioavailable P was stronger than the positive one mediated by microbial P. I think the focus here on the positive effect mediated by microbial processes is somewhat biased.
- Line 129. Several interlinked cause-effect relationships are described here with a reference to Fig. S1, which is a simple cross-correlation table. This really needs to be toned down significantly. Furthermore, "organic matter" is not the same as "organic carbon", and you only measured the latter if I understand correctly.
- Line 133. "feedback" is a noun, not a verb. Change to "feed back".
- Line 142. "factors" could be changed to "soil factors" for improved accuracy.
- Line 143. I have some questions regarding this SEM. First, what is the purpose of the "weather" composite variable as it only consists of a single measured variable, namely rainfall? Second, why do you assume that weather has no direct link with the biotic or abiotic soil factors? Why would management only affect plant species richness and bioavailable Pi, and not the biotic and abiotic soil factors? Why would there be no direct link between abiotic and biotic soil factors? Many logically defensible paths are not included in the model.
- Line 150. I have read "dual role" a few times now and it is still not clear what this refers to. Please define each role more clearly.
- Line 152. Is it not rather self-evident that fixing a path will result in a poorer model fit than when allowing all paths to be estimated freely? I fail to see how you can deduce any ecological meaning from this statistical self-evidence.
- Line 154. A good example of the sort of vague and woolly wording that occur throughout the manuscript: "plant species richness acts as a driver of environmental conditions". This is obviously true, but you only investigated its potential role as a driver of bioavailable inorganic phosphorus,

so I would just call it that.

- Line 160. Is this just a possible explanation, or is this based on any of the results? I presume the former, but then a reference to literature backing this up as a plausible mechanism would be good here.

- Line 166. I find "mathematically minimized" a little misleading. To me it suggests that you performed variation partitioning of P cycle variables into pure and shared effects of plant diversity and management. This would actually be a good way to go about this. My point is that "minimized" suggests a mathematical minimum to me. Here, you reduced the effect of management under the assumption that total P pools strongly depend on management. I can agree with that, yet management effects undoubtedly manifest themselves in other ways than by their influence on total P pools, and knowing these other ways would enable you to even further reduce the management effect here, so what you did does not necessarily represent a minimum. I would use "reduce" or something along those lines instead.

- Lines 180 - 183 are stated as fact while they are rather speculative. The authors jump from describing correlations in the previous lines to several mentions of "effects" here. Let me offer an alternative explanation, namely that species-rich calcareous grasslands are dominated by entirely different plant species than nutrient-rich grasslands for various reasons, including being thermophilic, drought-tolerant, or calcicole. I should probably add grazing-tolerant as nearly all European calcareous grasslands can only exist because of regular grazing or mowing, apart for the very driest xeroseres. I would argue that more efficient resource use in calcareous grassland plant assemblages has little to do with their high diversity as such, but with the fact that these specific plant species need a high resource use efficiency to cope with the harsh conditions of their habitat. High species richness is likely a consequence of the fact that in this harsh environment no single plant species can attain dominance as happens in more nutrient-rich systems. Here, the authors suggest that high diversity in calcareous grasslands is a driver of resource use efficiency, while I would argue it is the other way around. In short, plant diversity effects in calcareous grasslands versus in agricultural fields is a bit of an apples-oranges type comparison. My real point though is that we can speculate all we want here, but a bunch of pairwise correlations are not going to generate much mechanistic insight into the matter.

- Line 186. "was achieved": stating speculation as fact again.

- Line 187. "more species-rich community of AMF" ... "at high plant species richness" but according to Fig. S4, the correlation between plant species richness and AMF diversity was not significant, so this is not very compelling.

- Line 191. I assume the reference to Fig. 3 is to Fig. 3c more specifically here. I have no idea where these triangles and rectangles in Fig. 3c come from or on what data they are based, as no units or axes are present. Please clarify.

- Line 196. Alongside what?

- Line 200. I find the word "evidence" quite an overstatement here. After reading this whole paragraph and the many different multivariate hypotheses put forward in it, which are only supported by selected bivariate correlations, I find it surprising that the authors did not opt for SEM here to seek further validation of these explanations, especially given their use of SEM in other parts of the manuscript. While still far from enabling the generation of "evidence" it would likely enable the authors to retain or discard some of the hypotheses put forward in this paragraph.

- Line 207. A "unified theory" is an even bigger stretch given the data and results presented in this manuscript.

- Line 209. So Fig. 3a only holds for nutrient-rich soils then?

- Line 217. The proposed mechanisms here seem to make sense based on the results, but why would you not just recommend experiments manipulating both biodiversity and soil nutrient availability in a crossed design rather than separate biodiversity experiments under nutrient-poor conditions? They will still differ in many aspects of experiments in nutrient-rich systems. Would the ideal not be to have a single experiment, keeping all else constant, that varies all drivers of interest in a fully-crossed manner? The cited study by Craven et al. would also be relevant here.

- Line 406. Why are these averages considered more "robust"? I would argue that by averaging, information is lost. Why not use both years' data in a single model with year as an effect (random or fixed)?

- Line 440. "before the growing season started i.e., in September 2013". I would expect that in Germany the growing season should be nearing its end in September.

- Line 544. Pearson correlations come with the assumption of data independence. The data from

the agricultural fields violate this assumption, as they were obtained from plots with a clear spatial clustering as some plots stem from Schwäbische Alb and others from Hainich-Dün. It is likely that within each study region plot locations show further clustering. This should be taken into account to avoid overestimating effective degrees of freedom and hence having artificially low P-values. I would recommend mixed models with at the very least study region included as a random effect. Better would be to explicitly test all model residuals for spatial autocorrelation at various spatial scales and if significant autocorrelation is detected, to model this spatial autocorrelation in the dependent variables via the variance-covariance matrix, for example with an autoregressive or exponential distance decay depending on the shape of the observed autocorrelation. On a separate note, with about 300 tests, about 15 of the significant ones are probably so purely by chance. Multiple testing correction seems in order here, and I would recommend False Discovery Rate as described in, for example: Pike. 2011. *Methods in Ecology and Evolution* 2: 278-282.

- Line 557. Here is a good definition of the dual role that could have been made explicit upon first mentioning it: "as driving and driven by P availability".
- Line 567. The authors regularly cite Grace. This, too, is a quote from Grace: "Results based on a modified model must be considered provisional. Thus, it is not proper to present results of an analysis based on a modified model and claim that the resulting model has been adequately evaluated. When a single data set is used, only the initial model is subjected to a confirmatory assessment, the modified model is actually "fitted" to the data. Where sufficient samples are available, multiple evaluations can be conducted by splitting a data set, with each half being examined separately". I would strongly advise the authors to interpret their model accordingly.
- Line 706. Strange wording. The Pi concentrations were not adequately predicted by the SEM. I do not see how a variable can meaningfully deviate from a model.

Reviewer #2 (Remarks to the Author):

This work employed both grassland biodiversity experiment and agricultural grasslands to investigate the role of biodiversity (both of aboveground and below ground) in the P cycle. This topic is very interesting, and the MS is well written. The authors not only clearly described and discussed the results they obtained, but also attempted to put their biodiversity theory into practice. However, some issues should still be addressed:

1, The title is "Soil fertility modulates biodiversity – P cycle relationships". Regarding to the Soil fertility, how did you evaluate the soil fertility as the authors did not quantify the soil fertility. The grasslands, subjected to intense and less intensive managements, can not simply be classed into fertile or less fertile soils, respectively. Maybe the authors can calculate the soil fertility index to quantify the Soil fertility [Mukherjee A, Lal R (2014) Comparison of Soil Quality Index Using Three Methods. *PLOS ONE* 9(8): e105981]. All the SEMs, I cannot find the parameters to represent the soil fertility. Maybe only Corg? That' not clear.

2, the work aimed to explore mechanisms underlying the biodiversity – P cycle relationships. However, the authors did not well address the P cycle, but mainly focused on the Pi bioavailability. Yes, the Pi bioavailability is very important, but only a part of P cycle. The work omitted some other processes evolved in P cycle, which may be also very important for P retention and activation.

3, It is well established that the AMF is very important in mobilizing P. Referring to P, any scientist should think of AMF, which is not novel. So, it is not accidental that the more species-rich community of AMF increased microbial P exploitation (Fig. 3b). The readers should be more interested in the role of other microbes in P mobilization. For example, the phosphomonoesterase-secreting microbes. Under the current technique, investigation of the PAsE-secreting microbes based on genes (e.g. *phoD* and so on) is available.

4, In the SEMs or for testing the priori model, why the AMF biomass, abundance and richness did not take into account? Except for the AMF richness, the AMF biomass should also have a positive correlation with the P exploitation.

Some other comments:

1, Line 83: ".....mediated by plant-microbe interactions". In your data, which characterized the interactions?

2, Line 145-147: this sentence is not clear, and easily to make readers confused. Please refine.

3, Line 191, 194: abundance or relative abundance? How do you determine the AMF absolute abundance?

Or even it is relative abundance, as you primer set largely targeted to AMF, not to overall fungi, the relative abundance should not be convincing.

4, Line 393-394: "a gradient of management intensity with different fertilization". How do you quantify the gradient of fertilization? Do you take mineral NPK into account individually or integratively?

I also noticed that the Fertilization variable used in SEM is transformed by square root (Table S1). How do you value the fertilization and make it transformed.

5, Line 533: the reference for the primer set must be included.

6, Line 538: Only those AMF appearing on more than five plots were considered? Why? Some species, with high abundance but observed in a few places, should be categorized as specialists, which could still play profound role in the plot.

7, Fig. 2b, what is the meaning of the black arrow?

8, Table S1, how do you determine Soil fungi : bacteria ratio?

Reviewer #3 (Remarks to the Author):

The paper addresses timely and important question and does so well. I find the research well executed, presented and framed, and questions well formulated. Thus I have only minor technical suggestions towards further improving the paper.

The only major aspect related to the data availability. Namely, DNA sequence data (such as fungal sequences generated in the paper) are commonly to the research area submitted to a public DNA sequence repository. This manuscript states that the data will be made available as supplementary files and on project web pages upon paper acceptance. This would not do favor to the usability of the painstakingly gathered data as these sequences would not be searchable and easy to incorporate in biodiversity queries and further analyses.

minor comments:

L76: mycorrhizal fungal species

L82: mycorrhizal fungal diversity. please note that mycorrhizal diversity would refer to the diversity of mycorrhizal types (e.g., co-occurrence of arbuscular, ecto- etc mycorrhizal associations in a system)

L117: biodiversity effects on P cycle - is the directionality always clear?

L527 and 529: the data on fungal diversity from the two systems (biodiversity experiment and agricultural grasslands) stem from the two different sampling years. For some readers this would be a complication, as the sampling time coincides with the study system. though AM fungal diversity in study systems is not compared in this study, would a brief comment be worthwhile to which degree such aspect might contribute to the data variability of this analysis, and, potentially, to the conclusions drawn

L533: please provide references to the primers (original papers describing these primers)

L534: which Illumina MiSeq version? 2x300 bp?

L536: reference needed for the Mothur software

L537: reference needed for the MaarjAM database

Response to reviewer comments

Reviewer #1 (Remarks to the Author):

This study tries to disentangle the complex relationships among soil nutrient status, plant diversity, and P cycling. This is a very relevant topic, and as the authors highlight, a better mechanistic understanding of the drivers affecting the link between plant diversity and P availability may hold important consequences for the optimization of agricultural practices both in terms of biodiversity conservation and fertilizer reduction. I was therefore quite excited reading the title of the manuscript. However, after reading it, I was left with the impression that the data obtained and analyses performed, while commendable in many ways, did not justify the strength of the claim made by the title.

Response: We thank the referee for underlining the relevance of our work and for the constructive feedback. We modified the title. We believe that the results of the re-analysis now fully back the title.

I see several issues with this study, some fundamental in nature and others more practical. First, the only manipulated variable across all plots studied is plant diversity. Overall nutrient richness or P availability were never manipulated. This is not a criticism, as I very well understand the difficulty of manipulating multiple factors especially in a large-scale field setting, but an observation. It necessarily implies that any relationships linking plant diversity on the one hand to ultimate response variables on the other, are essentially correlative. There is nothing wrong with correlative work, and I believe SEM is the ideal tool to explore hypotheses about the modulating influences of intermediate variables. However, care should be taken in claiming causality based on such correlative work. A step in the right direction is applying a hypothetical model based on one set of data to a separate data set for validation, as the authors did. In this case, however, the model performed poorly on the new data. The authors' response was to build a new model including a factor (management) they thought could explain the discrepancy. This is a good next step, but unfortunately, the authors did not proceed by testing their new, now better fitting model including management on a separate validation data set as they did with their first model but seemed to consider this model "final". The authors nonetheless explain many of the patterns they see in their data by the effect of management, which was poorly validated and again, not a manipulated factor. In other words, the final model based on the agricultural plots is entirely correlative and fitted to the data rather than confirmatory. I therefore urge the authors to treat it as such. This means using much more care in drawing conclusions from these models, which are exploratory at best.

Response: We fully agree that the procedure described by the reviewer represents the ideal way to tease apart effects of management and plant diversity. We tried to split the data set of the agricultural grasslands into two (by region or randomly), but the sample size prevented the convergence of the models. Therefore, regrettably, there is no reliable data set for validation of the model for the established grasslands available. Yet, this work can provide the basis for testing/validating the model by other researchers. Overall, we checked our conclusions and toned down the statements accordingly.

In addition to these rather fundamental comments, I have a few issues regarding the way in which data were analyzed. I refer to my detailed comments below but summarize the essence here. First, in the SEM of the agricultural grasslands, a case could be made for several logical pathways that were not included. If there were valid reasons for not including these, they should be given and explained.

Response: We changed the model for the agricultural grasslands. All relevant paths are now included. Refer as well to our response to I. 143.

Second, I do not see why many of the conclusions are drawn from the table of rough pairwise Pearson correlations provided in Fig. S4., especially about variables mediating relationships among other variables, for which (another) SEM would make more sense.

Response: The new models now comprise all relevant paths, and we rely on these to draw conclusions.

Third, for the experimental plots, which were likely randomized in space, this is less of an issue, but the plots in the agricultural fields are clearly not independent, as they are clustered (likely at multiple levels) in space. Violating the assumption of independent data will lead to erroneous conclusions regarding any analyses in which the dependence structure is not properly taken into account. To my understanding, this was done neither in the SEM neither in the Pearson correlations.

Response: As suggested by the editor, we now used the “piecewise SEM” package in R to account for spatial clustering of the plots within the two regions of the agricultural grasslands. We further tested for spatial clustering within each region and can now exclude spatial autocorrelation. Please refer to our detailed response to your comment on l. 544.

Fourth, running a very large number of significance tests on Pearson correlations will result in about 5% of them turning out significant purely by chance. I know there is quite some discussion about the merits of multiple testing corrections, but in this case it seems advisable.

Response: We thank the reviewer for this advice. In the revised manuscript, we now report the significance of Pearson correlations based on adjusted p-values based on the Benjamini-Hochberg Method (Benjamini & Hochberg 1995).

Finally, I found the writing often a tad overgeneralizing and vague. For further details I refer to my individual comments listed below.

Response: We checked our writing style and clarified vague statements.

Detailed comments:

- The abstract is confusing without reading the remainder of the manuscript, yet should be understandable as a self-contained piece of text. Roughly summarized, it states that a) plant biodiversity stimulates P cycling in nutrient-rich soils, b) P availability has a negative effect on plant biodiversity. This immediately begs the question what is meant by nutrient-rich. P is also a nutrient. If “nutrients” are good for the plant diversity - P cycle relationship but P itself is bad, does this mean “nutrients other than P”? Then the sentence “Our results demonstrate that promoting a high plant biodiversity on nutrient-rich soils and a high soil microbe, particularly AMF, biodiversity on nutrient-poor soils has ecological (biodiversity protection) and economical (fertiliser savings) benefits.” It is not clear from the preceding sentences why promoting plant biodiversity has to happen on nutrient-rich soils and promoting AMF on nutrient-poor soils and not the other way around. I suggest you rephrase the abstract to ensure that each sentence logically builds on previous statements.

Response: We rewrote the abstract making sure that we stuck to the logical suite of thoughts. Because of the slightly changed outcome of the new models, the fertility argument is no longer relevant.

- Line 57. “profits” should be “profit” as a mass noun, or possibly “benefits”

Response: Changed as suggested.

- Line 68. I suggest changing “nutrient” to “nitrogen” here. Otherwise the “other nutrients” in the next sentence makes little sense.

Response: Sentence deleted.

- Line 70. I would think terrestrial ecologists have been aware of the critical role of P for quite a long time, so I am not sure whether there is still a “growing awareness”.

Response: „The reviewer is right. Therefore, we have replaced “awareness” (which was indeed high for a long time) by “importance as a limiting nutrient” and have added the reasons for this growing importance, i.e. high N deposition and decreasing availability of mineral P fertilisers (revised version l. 74-76).

- Line 76. I suggest changing “mycorrhizal species” to “mycorrhizal fungi species”.

Response: Changed as suggested.

- Line 77-79: This sentence lacks some clarity. Does “symbiotic partner” still refer to AMF here, or to the wider soil microbial community? Please clarify.

Response: “symbiotic partner” replaced by “microbial community”.

- Line 92. “two way” should be “two-way”

Response: Sentence deleted, comment does no longer apply.

- Line 96. Filter of what? Of plant species composition? Please clarify.

Response: Clarified: “of the community composition of plants and soil microorganisms”.

- Line 98. Not only small-scale, I would argue.

Response: “small-scale” deleted.

- Line 100. This sentence is ambiguous too, as “different directions” can refer to a difference among filters or a difference between effects on plants and effects on soil organisms. I assume the latter is meant here, but please rephrase for clarity.

Response: Sentence deleted, comment does not longer apply.

- Line 102. The meaning of this sentence also evades me. How does the fact that mechanisms may have been overlooked follow from the fact that prior positive biodiversity effects on productivity have been reported? What is the logic here? It reads as if you imply that scientists only look for mechanisms of negative effects and not of positive ones.

Response: Sentence deleted, comment does not longer apply.

- Line 128. I would advise the authors to be careful in their wording regarding SEMs. I suggest changing “originated from” to “related to”. Also, the total path strength of plant species richness to bioavailable P mediated by organic C was 0.35. The total path via microbial P had a coefficient of only 0.14. The negative path via phosphatase activity in the soil had a coefficient of -0.15. The negative link between plant species richness and bioavailable P was stronger than the positive one mediated by microbial P. I think the focus here on the positive effect mediated by microbial processes is somewhat biased.

Response: The paths itself as well as the path strengths changed because we used the ‘piecewise SEM’ package and restructured the model. Nevertheless, we carefully phrased the descriptions of the new models following your advice.

- Line 129. Several interlinked cause-effect relationships are described here with a reference to Fig. S1, which is a simple cross-correlation table. This really needs to be toned down significantly.

Furthermore, “organic matter” is not the same as “organic carbon”, and you only measured the latter if I understand correctly.

Response: We toned down all statements that related to bivariate correlations. We now correctly use the term organic carbon.

- Line 133. “feedback” is a noun, not a verb. Change to “feed back”.

Response: Sentence deleted, comment does no longer apply.

- Line 142. "factors" could be changed to "soil factors" for improved accuracy.

Response: Sentence deleted, comment does no longer apply.

- Line 143. I have some questions regarding this SEM. First, what is the purpose of the "weather" composite variable as it only consists of a single measured variable, namely rainfall? Second, why do you assume that weather has no direct link with the biotic or abiotic soil factors? Why would management only affect plant species richness and bioavailable Pi, and not the biotic and abiotic soil factors? Why would there be no direct link between abiotic and biotic soil factors? Many logically defensible paths are not included in the model.

Response: As suggested by you and by the editor, we used another type of SEM and restructured the model, so that this comment does no longer apply.

- Line 150. I have read "dual role" a few times now and it is still not clear what this refers to. Please define each role more clearly.

Response: The new models do not explicitly consider the dual role of plant species (plant species as a driver of processes as evident from biodiversity experiments [plant species richness = independent variable] and plant species richness as a response to processes [plant species richness = dependent variable]), and we have removed the term "dual role". Therefore, this comment does no longer apply.

- Line 152. Is it not rather self-evident that fixing a path will result in a poorer model fit than when allowing all paths to be estimated freely? I fail to see how you can deduce any ecological meaning from this statistical self-evidence.

Response: Because we used another type of SEM, this comment does no longer apply.

- Line 154. A good example of the sort of vague and woolly wording that occurs throughout the manuscript: "plant species richness acts as a driver of environmental conditions". This is obviously true, but you only investigated its potential role as a driver of bioavailable inorganic phosphorus, so I would just call it that.

Response: We checked the manuscript for vague statements and replaced them by precise ones.

- Line 160. Is this just a possible explanation, or is this based on any of the results? I presume the former, but then a reference to literature backing this up as a plausible mechanism would be good here.

Response: Sentence deleted, comment does not longer apply.

- Line 166. I find "mathematically minimized" a little misleading. To me it suggests that you performed variation partitioning of P cycle variables into pure and shared effects of plant diversity and management. This would actually be a good way to go about this. My point is that "minimized" suggests a mathematical minimum to me. Here, you reduced the effect of management under the assumption that total P pools strongly depend on management. I can agree with that, yet management effects undoubtedly manifest themselves in other ways than by their influence on total P pools, and knowing these other ways would enable you to even further reduce the management effect here, so what you did does not necessarily represent a minimum. I would use "reduce" or something along those lines instead.

Response: We now show P exploitation for the biodiversity experiment as well where the management was not different across plant species richness levels. Therefore, we no longer refer to the minimization/reduction of the management effect regarding P exploitation.

- Lines 180 - 183 are stated as fact while they are rather speculative. The authors jump from describing correlations in the previous lines to several mentions of "effects" here. Let me offer an alternative explanation, namely that species-rich calcareous grasslands are dominated by entirely

different plant species than nutrient-rich grasslands for various reasons, including being thermophilic, drought-tolerant, or calcicole. I should probably add grazing-tolerant as nearly all European calcareous grasslands can only exist because of regular grazing or mowing, apart for the very driest xeroseres. I would argue that more efficient resource use in calcareous grassland plant assemblages has little to do with their high diversity as such, but with the fact that these specific plant species need a high resource use efficiency to cope with the harsh conditions of their habitat. High species richness is likely a consequence of the fact that in this harsh environment no single plant species can attain dominance as happens in more nutrient-rich systems. Here, the authors suggest that high diversity in calcareous grasslands is a driver of resource use efficiency, while I would argue it is the other way around. In short, plant diversity effects in calcareous grasslands versus in agricultural fields is a bit of an apples-oranges type comparison. My real point though is that we can speculate all we want here, but a bunch of pairwise correlations are not going to generate much mechanistic insight into the matter.

Response: Thank you for your valuable comments. Our comparison includes calcareous grasslands only (no non-calcareous agricultural fields considered). Yet, the species-rich plant communities in our study are managed at a low intensity likely because of the special site conditions (e.g., dryness) that you term “harsh” from a plant perspective. Because we cannot resolve cause and effect here, we refrain from the argument focusing on resource use efficiency.

- Line 186. “was achieved”: stating speculation as fact again.

Response: Rephrased.

- Line 187. “more species-rich community of AMF” ... “at high plant species richness” but according to Fig. S4, the correlation between plant species richness and AMF diversity was not significant, so this is not very compelling.

Response: We now considered AMF in the new SEMs and adjusted the statements accordingly.

- Line 191. I assume the reference to Fig. 3 is to Fig. 3c more specifically here. I have no idea where these triangles and rectangles in Fig. 3c come from or on what data they are based, as no units or axes are present. Please clarify.

Response: Figure deleted, comment does no longer apply.

- Line 196. Alongside what?

Response: Sentence deleted, comment does no longer apply.

- Line 200. I find the word “evidence” quite an overstatement here. After reading this whole paragraph and the many different multivariate hypotheses put forward in it, which are only supported by selected bivariate correlations, I find it surprising that the authors did not opt for SEM here to seek further validation of these explanations, especially given their use of SEM in other parts of the manuscript. While still far from enabling the generation of “evidence” it would likely enable the authors to retain or discard some of the hypotheses put forward in this paragraph.

Response: Paragraph deleted because of focus on P exploitation in the revised version.

- Line 207. A “unified theory” is an even bigger stretch given the data and results presented in this manuscript.

Response: Paragraph deleted because of focus on P exploitation in the revised version.

- Line 209. So Fig. 3a only holds for nutrient-rich soils then?

Response: Figure and paragraph deleted because of focus on P exploitation in the revised version.

- Line 217. The proposed mechanisms here seem to make sense based on the results, but why would you not just recommend experiments manipulating both biodiversity and soil nutrient availability in a

crossed design rather than separate biodiversity experiments under nutrient-poor conditions? They will still differ in many aspects of experiments in nutrient-rich systems. Would the ideal not be to have a single experiment, keeping all else constant, that varies all drivers of interest in a fully-crossed manner? The cited study by Craven et al. would also be relevant here.

Response: We agree in part. Theoretically, an experiment would be ideal to test the role of all drivers of interest including interactions among different drivers. Actually, such experiments have been conducted and yielded important insights (e.g., Reich et al. 2001; Fridley 2002; Craven et al. 2016). Yet, such approaches are limited because it is not possible to realize a fully crossed design given the dependencies among drivers. For example, a high species richness is incompatible with high nutrient availability and accordingly, high nutrient availability levels will result in low plant species richness. Because of constraints of both experimental and observational approaches, we are convinced that the combination of the two approaches is a promising avenue to explore whether or not experimental evidence plays out in ecosystems. We now further stress this issue in the introduction to justify of our approach (revised version l. 112-115).

- Line 406. Why are these averages considered more “robust”? I would argue that by averaging, information is lost. Why not use both years’ data in a single model with year as an effect (random or fixed)?

Response: The P cycle responds to changing conditions with some delay because of the equilibrium among different P pools in soil which is partly only reached with a slow kinetics (Helfenstein et al. 2018, Nature Communications 9: 9). Consequently, management effects could manifest year(s) later. Therefore, we do not regard relations among variables just within one year to be reliable.

- Line 440. “before the growing season started i.e., in September 2013”. I would expect that in Germany the growing season should be nearing its end in September.

Response: Rephrased (no mention of growing season anymore).

- Line 544. Pearson correlations come with the assumption of data independence. The data from the agricultural fields violate this assumption, as they were obtained from plots with a clear spatial clustering as some plots stem from Schwäbische Alb and others from Hainich-Dün. It is likely that within each study region plot locations show further clustering. This should be taken into account to avoid overestimating effective degrees of freedom and hence having artificially low P-values. I would recommend mixed models with at the very least study region included as a random effect. Better would be to explicitly test all model residuals for spatial autocorrelation at various spatial scales and if significant autocorrelation is detected, to model this spatial autocorrelation in the dependent variables via the variance-covariance matrix, for example with an autoregressive or exponential distance decay depending on the shape of the observed autocorrelation. On a separate note, with about 300 tests, about 15 of the significant ones are probably so purely by chance. Multiple testing correction seems in order here, and I would recommend False Discovery Rate as described in, for example: Pike. 2011. Methods in Ecology and Evolution 2: 278-282.

Response: We thank the reviewer for this valuable comment. In the revised version, we accounted for the spatial clustering in the SEMs using the R package “piecewise SEM”. This package allowed us to fit “region” as random intercept term in mixed effects models and accounted for the spatial clustering of the plots within the different regions. Furthermore, we checked for spatial autocorrelation within each region (example in Response Fig. 1).

Response Figure 1: Scatterplots of the residuals of the total P exploitation model from the SEM to show the spatial dependence within the regions. The residuals are plotted separately for each region (Schwäbische Alb = AEG, Hainich = HEG). Shown are the residuals vs. easting (RW) and vs. northing (HW).

To this end, we related the residuals associated to each variable in the SEM to the geographic coordinates of each plot. Since we found no indications of relationships or clustering of the residuals, we assumed independent data within the regions of the agricultural grasslands. We added this statement in the Methods. Finally, in the revised version the p-values of the Pearson correlations are corrected for multiple inference using the Benjamini-Hochberg correction.

- Line 557. Here is a good definition of the dual role that could have been made explicit upon first mentioning it: “as driving and driven by P availability”.

Response: Because the dual role of plant species richness cannot be tackled as explicitly as in the former manuscript version in which the SEMs comprised composite variables, we shifted the focus to the biodiversity – management contrast in the revised version. Therefore, the definition is no longer needed.

- Line 567. The authors regularly cite Grace. This, too, is a quote from Grace: “Results based on a modified model must be considered provisional. Thus, it is not proper to present results of an analysis based on a modified model and claim that the resulting model has been adequately evaluated. When a single data set is used, only the initial model is subjected to a confirmatory assessment, the modified model is actually “fitted” to the data. Where sufficient samples are

available, multiple evaluations can be conducted by splitting a data set, with each half being examined separately". I would strongly advise the authors to interpret their model accordingly.

Response: Please refer to our more detailed answer to the first general comment of the reviewer ("First, the only manipulated variable across all plots studied is plant diversity.").

- Line 706. Strange wording. The Pi concentrations were not adequately predicted by the SEM. I do not see how a variable can meaningfully deviate from a model.

Response: Figure and caption deleted because of the focus on P exploitation in the revised version.

Reviewer #2 (Remarks to the Author):

This work employed both grassland biodiversity experiment and agricultural grasslands to investigate the role of biodiversity (both of aboveground and below ground) in the P cycle. This topic is very interesting, and the MS is well written. The authors not only clearly described and discussed the results they obtained, but also attempted to put their biodiversity theory into practice. However, some issues should still be addressed:

1, The title is "Soil fertility modulates biodiversity – P cycle relationships". Regarding to the Soil fertility, how did you evaluate the soil fertility as the authors did not quantify the soil fertility. The grasslands, subjected to intense and less intensive managements, can not simply be classed into fertile or less fertile soils, respectively. Maybe the authors can calculate the soil fertility index to quantify the Soil fertility [Mukherjee A, Lal R (2014) Comparison of Soil Quality Index Using Three Methods. PLOS ONE 9(8): e105981]. All the SEMs, I cannot find the parameters to represent the soil fertility. Maybe only Corg? That' not clear.

Response: Thank you for your valuable comments. This particular issue was also mentioned by Reviewer #1. We rephrased the title by focusing on the P cycle.

2, the work aimed to explore mechanisms underlying the biodiversity – P cycle relationships. However, the authors did not well address the P cycle, but mainly focused on the Pi bioavailability. Yes, the Pi bioavailability is very important, but only a part of P cycle. The work omitted some other processes evolved in P cycle, which may be also very important for P retention and activation.

Response: Agreed. We now refocused the manuscript on P exploitation and thereby included aspects of the P cycle beyond bioavailable P.

3, It is well established that the AMF is very important in mobilizing P. Referring to P, any scientist should think of AMF, which is not novel. So, it is not accidental that the more species-rich community of AMF increased microbial P exploitation (Fig. 3b). The readers should be more interested in the role of other microbes in P mobilization. For example, the phosphomonoesterase-secreting microbes. Under the current technique, investigation of the PAsE-secreting microbes based on genes (e.g. phoD and so on) is available.

Response: Thank you for your valuable comment. Indeed, we found that not only AMF but also other microbes/the microbial community play a role for P mobilization. We state this explicitly in the main text (l. 167-170). Regarding the second part of the comment, we did not focus on functional genes but on taxonomic units. Indeed, given the availability of lists of taxa who produce the PAsE it would be interesting to check how abundant these were in our dataset, provided sequencing depth was deep enough. However, this approach would go beyond the scope of our study and rather represent the starting point for a follow up study. Nevertheless, we now included this valuable idea as an outlook in our discussion (l. 201-203).

4, In the SEMs or for testing the priori model, why the AMF biomass, abundance and richness did not take into account? Except for the AMF richness, the AMF biomass should also have a positive correlation with the P exploitation.

Response: Following your advice and that of Reviewer #1 and of the editor, we set up new SEMs which now include AMF richness and abundance.

Some other comments:

1, Line 83: “.....mediated by plant-microbe interactions”. In your data, which characterized the interactions?

Response: The SEMs provide the theory-based links among variables and include plant- and soil microorganism-related interactions (see Table S1 and Figs. S1 and S2).

2, Line 145-147: this sentence is not clear, and easily to make readers confused. Please refine.

Response: Sentence deleted, comment does no longer apply.

3, Line 191, 194: abundance or relative abundance? How do you determine the AMF absolute abundance? Or even it is relative abundance, as you primer set largely targeted to AMF, not to overall fungi, the relative abundance should not be convincible.

Response: We refer to the relative abundance (see methods). We now replaced “abundance” by “relative abundance” in the text.

4, Line 393-394: “a gradient of management intensity with different fertilization”. How do you quantify the gradient of fertilization? Do you take mineral NPK into account individually or integratively?

I also noticed that the Fertilization variable used in SEM is transformed by square root (Table S1). How do you value the fertilization and make it transformed.

Response: The amount of organic and inorganic fertilizer applied annually is based on information received from all farmers involved in the project. All fertilizer sources are summed and thus, we follow an integrative approach. We now rely on an index of management intensity combining fertilization, grazing, and mowing intensity (“LUI”). Therefore, no further transformation is necessary. For further information refer to Blüthgen et al. 2012; Basic and Applied Ecology 13: 207-220.

5, Line 533: the reference for the primer set must be included.

Response: Sorry for this lapse, we now included the references for the primer sets.

6, Line 538: Only those AMF appearing on more than five plots were considered? Why? Some species, with high abundance but observed in a few places, should be categorized as specialists, which could still play profound role in the plot.

Response: Those AMF appearing on less than five plots had low relative abundances (< 1%) and thus, were considered to play a negligible role. We added this information in the methods.

7, Fig. 2b, what is the meaning of the black arrow?

Response: Figure deleted, comment does no longer apply.

8, Table S1, how do you determine Soil fungi : bacteria ratio?

Response: We thank the reviewer for pointing this out. The fungal to bacterial ratios were calculated on the basis of PLFAs. We added this information in the methods.

Reviewer #3 (Remarks to the Author):

The paper addresses timely and important question and does so well. I find the research well executed, presented and framed, and questions well formulated. Thus I have only minor technical suggestions towards further improving the paper.

The only major aspect related to the data availability. Namely, DNA sequence data (such as fungal sequences generated in the paper) are commonly to the research area submitted to a public DNA sequence repository. This manuscript states that the data will be made available as supplementary files and on project web pages upon paper acceptance. This would not do favor to the usability of the painstakingly gathered data as these sequences would not be searchable and easy to incorporate in biodiversity queries and further analyses.

Response: A part of the data (biodiversity experiment) is available on Pangea already (<https://doi.pangaea.de/10.1594/PANGAEA.874990>).

minor comments:

L76: mycorrhizal fungal species

Response: Rephrased as suggested.

L82: mycorrhizal fungal diversity. please note that mycorrhizal diversity would refer to the diversity of mycorrhizal types (e.g., co-occurrence of arbuscular, ecto- etc mycorrhizal associations in a system)

Response: We rephrased and now use the term “AMF species richness” throughout the manuscript.

L117: biodiversity effects on P cycle - is the directionality always clear?

Response: Paragraph deleted because of focus on P exploitation in revised version.

L527 and 529: the data on fungal diversity from the two systems (biodiversity experiment and agricultural grasslands) stem from the two different sampling years. For some readers this would be a complication, as the sampling time coincides with the study system. though AM fungal diversity in study systems is not compared in this study, would a brief comment be worthwhile to which degree such aspect might contribute to the data variability of this analysis, and, potentially, to the conclusions drawn

Response: Our SEM models explained nearly half of the total variance in the data (32 and 34% of P exploitation in the agricultural grasslands and in the biodiversity experiment, respectively, plus 10% explained variance by the random factor region [Table S5]). The temporal variability might contribute to the unexplained remainder. However, given that the variables that were included in the SEMs explained nearly half of the total variance, we are confident that the conclusions we draw (toned down as suggested) are reliable.

L533: please provide references to the primers (original papers describing these primers)

Response: Sorry for this lapse, we now included the references for the primer sets.

L534: which Illumina MiSeq version? 2x300 bp?

Response: We now included this information.

L536: reference needed for the Mothur software

Response: We now included this information.

L537: reference needed for the MaarjAM database

Response: We now included this information.

References

Benjamini Y, Hochberg Y. Controlling the false discovery rate - a practical and powerful approach to multiple testing. *J R Stat Soc Ser B-Stat Methodol* 57, 289-300 (1995).

Craven D, et al. Plant diversity effects on grassland productivity are robust to both nutrient enrichment and drought. *Phil Trans R Soc B* 371, 8 (2016).

Fridley JD. Resource availability dominates and alters the relationship between species diversity and ecosystem productivity in experimental plant communities. *Oecologia* 132, 271-277 (2002).

Reich PB, et al. Plant diversity enhances ecosystem responses to elevated CO₂ and nitrogen deposition. *Nature* 410, 809-812 (2001).

Reviewer comments, second round –

Reviewer #1 (Remarks to the Author):

Let me first apologize for my delay in submitting this review. I believe the authors did a fine job improving their manuscript based on the previous comments and I have no prohibitive issues with the new manuscript at this point. I appreciate that the authors took the effort to reanalyze their data in a statistically sounder way, and I think Table S1 is a great addition. I still feel that here and there more cautious phrasing of results is in order and refer to my detailed comments below in that regard. Furthermore, the results section is a bit of a mix of describing SEM and then raw Pearson correlations, and it is not always clear why certain variables were included in the SEM and then others were merely correlated with these variables. For example, the relationship between plant richness and plant P stocks in the biodiversity SEM is explained in terms of plant biomass, but plant biomass is not in the SEM, so that the explanation is based on a rather coarse Pearson correlations (which does not take into account possible issues of spatial data dependency, see detailed comments below). I fully understand the difficulty of selecting meaningful variables a priori to build conceptual SEM, and again, Table S1 nicely sets out why these variables were chosen. Still, I would be careful not to attach too much value to these Pearson correlations, and treat them merely as possible explanations for some of the pathways in the SEM, admitting the speculative nature of these explanations.

Detailed comments (referring to line numbers in the file with changes highlighted):

- Line 70: Suggest changing “contrary of” to “contrary to”
- Line 71: Suggest changing to “However, P leaching was not reduced by increased plant species diversity, ...”
- Line 94: Suggestion: “... from an experimental perspective, biodiversity and management can be manipulated simultaneously.”
- Line 114. Here the true meaning of what you are trying to say only became clear after reading the response to the comment regarding line 217 in the previous manuscript, which I can concur with. However, I would clarify the sentence by adding a few more words, for example: “However, biodiversity and management are closely intertwined, with high nutrient concentrations sooner or later resulting in lower plant biodiversity, so that a fully crossed experiment cannot be maintained for very long in reality.”
- Line 116. “mechanistic understanding gained from experiments in observational data obtained from agricultural grasslands” is a lot to unpack. What do you mean by “mechanistic understanding”? I have the impression this is one of the most frequently misused phrases in current ecology. Furthermore, what are “experiments in observational data”? Either you do an experiment, i.e., you manipulate certain variables in order to test their effect on response variables, or you do an observational study, i.e., you study correlations among variables without manipulation. Both have their merits, but it is unclear to me what you mean here. Or do you mean something along the lines of “the combination of insights gained from both experiments and observational data”?
- Line 145. Suggest changing “increased by” to “positively related with”.
- Line 147. Suggest changing. “resulted” to “likely resulted”. This was not tested here.
- Line 149. I only find one measure in Table S2 related to organic C stocks: “Corg stocks”. How do I deduce from this table that “Corg stocks were closely related to microbial C stocks”, as the latter seems missing from the table?
- Line 150. Suggest changing “positively influenced” to “was positively correlated with”

- Line 151. Suggest changing “because microorganisms took up” to “as microorganisms typically take up”.
- Line 155. Suggest changing “Accordingly, a positive ...” to “Accordingly, our model indicates that a positive ...”.
- Line 161. Please indicate which result leads to the conclusion that plant C assimilation depends on plant species richness. I do not find anywhere that you measured plant C assimilation. If this is a general statement based on literature, please include references.
- Line 166. Suggest changing “feedback on” on to “feed back to” and “Neither was there” to “There was neither a ... nor a significant relationship between P concentrations or plant biomass and AMF species richness or relative abundance”.
- Line 173. Suggest changing “In conclusion, plant ...” to “In conclusion, our data suggest that ...”
- Line 179-182. Can be collapsed into a single sentence, by deleting “We explored the mode ... experiment”.
- Line 184. Regarding this “direct relationship”, you propose in Table S1 that the underlying mechanism is an increased abundance of exploitative plant species. Did you not have abundance data for these particular species? If so, this could have been added as an intermediate variable in the SEM. I am just wondering, because of course this added variable would complicate the SEM so that may be a reason to opt not to include it.
- Line 208. Plant biomass is not in Fig. 1B, so something is missing from this explanation. Do you mean that low management intensity increased plant diversity and that this in turn lowered P stocks because plant diversity is negatively correlated with plant biomass (as indicated by Table S3)? Please clarify.
- Line 211-214 are difficult to follow. Why is an effect on plant biomass “reversing the relationship between plant species richness and plant P stocks”? This is not at all self-evident. Basically, management changed the direction of the relationship between plant diversity and plant biomass from positive in the biodiversity experiment to negative in the agricultural grasslands (as supported by Table S2 and S3). My explanation would be that there is a unimodal relationship with biomass peaking at low diversity of a few dominant and highly competitive plant species, then dropping toward intermediate diversity and increasing again somewhat at high diversity. Then your agricultural grasslands would be on the lower-diversity part, i.e., the negative slope, and the biodiversity experiment on the higher-diversity part, with the positive slope. Please clarify.
- Line 230. But at the same time, reduced management intensity was bad for P exploitation if you look at the pathway via Corg stocks and Microbial P stocks. In such situations, calculating the combined effect may help shed light on which pathways are most important. Here, the net effect mediated through plant richness and AMF abundance is: $-0.64 \times 0.25 - 0.29 \times 0.31 = -0.25$. This is ignoring the non-significant paths for now, which actually should be taken into account as well. The net effect mediated through Corg and Microbial P stocks would then be $0.24 \times 0.54 \times 0.52 = 0.067$, which is indeed probably not significant, so I would agree with your conclusion. It may still help to mention that the latter effect is much smaller for clarity.
- Line 232. What I was missing after reading this paragraph is a possible explanation for the lack of significant relationship between Plant P stocks and plant P exploitation, which seems unexpected.
- Line 394. Based on Fig. S2, it seems like the paths from Corg stocks to P exploitation and from Plant richness to Plant P stocks were added afterwards and thus should also be depicted by double-lined arrows.
- Line 586. Let me copy my original remark and response of the authors: “Line 406. Why are these averages considered more “robust”? I would argue that by averaging, information is lost. Why not use both years’ data in a single model with year as an effect (random or fixed)?
Response: The P cycle responds to changing conditions with some delay because of the equilibrium

among different P pools in soil which is partly only reached with a slow kinetics (Helfenstein et al. 2018, Nature Communications 9: 9). Consequently, management effects could manifest year(s) later. Therefore, we do not regard relations among variables just within one year to be reliable.”

I fully agree that relations among variables just within one year are less reliable, that is why in my original comment I suggested to include values for both years, and include the year effect into the model, as this will preserve the variation among both years, which you lose when you average across both years. If you decide not to, it would be nice to see some argumentation as to why you think this temporal variation is not relevant in your study.

- Line 735. Statistical analyses. I much appreciate the changes made to the statistical analyses, which I think have improved significantly as a result. Still, I have a few smaller questions: 1) Your previous manuscript mentioned transformations of variables to improve residual normality and linearity of relationships. This is missing now from the new manuscript. I can imagine that as you used piecewise SEM, you now were able to fit generalized linear (mixed) models with data distributions better matching the actual data. If so, I think you should include a list of the individual models fitted, their data distribution (normal, Poisson, binomial etc.) and their link functions (identity, log, logit etc.). This could for example be added as a column to Table S1. 2) I personally would have generated spatial (spline) correlograms of residual correlation versus distance among sampling points, but I see no real issue with the approach of just visually inspecting plots of model residuals versus easting and northing. 3) It is great that you have adjusted the many Pearson correlations for multiple testing, but as I also said in my original comment, not just the linear models that make up the SEM, but also the Pearson correlations assume independence of collected data. In that sense, your p-values of the correlations, even after correcting for multiple testing, may still be artificially low because of overestimated numbers of independent samples per correlation. The only alternative would be linear models for these variables as well, similar to the ones in the SEM, and checking for spatial autocorrelation here too. I will not make this a make-or-break point for this manuscript, but just think you should be aware that you should interpret the correlation p-values in Table S2 and S3 with caution.

- Line 915. Table S1: I really appreciate the addition of this Table as it provides the necessary context for and reasoning behind the hypothesized pathways of the SEM. I just noticed the paths from microbial and plant P stocks to P exploitation are not mentioned here. I realize these may be rather self-evident, but it still would not hurt to include them here, even if the explanation of these two paths is somewhat self-explanatory.

- Line 920. PUE is not defined in the table caption. I would also define biomass as “plant biomass” for clarity, also in Table S3.

- Line 925. LUI is not defined in the table caption.

- Line 936. Table S5: Are these R-squared values from the final SEMs? Please clarify.

Reviewer #2 (Remarks to the Author):

Thank you for responding all my concerns. I principally satisfied with the answers.

Reviewer #1 (Remarks to the Author):

Let me first apologize for my delay in submitting this review. I believe the authors did a fine job improving their manuscript based on the previous comments and I have no prohibitive issues with the new manuscript at this point. I appreciate that the authors took the effort to reanalyze their data in a statistically sounder way, and I think Table S1 is a great addition. I still feel that here and there more cautious phrasing of results is in order and refer to my detailed comments below in that regard.

Response: Thank you for your thorough evaluation and the positive and constructive comments. We followed all suggestions including those on more cautious phrasing.

Furthermore, the results section is a bit of a mix of describing SEM and then raw Pearson correlations, and it is not always clear why certain variables were included in the SEM and then others were merely correlated with these variables. For example, the relationship between plant richness and plant P stocks in the biodiversity SEM is explained in terms of plant biomass, but plant biomass is not in the SEM, so that the explanation is based on a rather coarse Pearson correlations (which does not take into account possible issues of spatial data dependency, see detailed comments below). I fully understand the difficulty of selecting meaningful variables a priori to build conceptual SEM, and again, Table S1 nicely sets out why these variables were chosen. Still, I would be careful not to attach too much value to these Pearson correlations, and treat them merely as possible explanations for some of the pathways in the SEM, admitting the speculative nature of these explanations.

Response: We checked all references to the bivariate correlations in the main text. Overall, we reduced the reliance on bivariate correlations (deleted in lines 148, 149). We kept only those that have been elucidated already in more detail in other studies (literature references added in lines 148, 149, 156-159). Nevertheless, we also adjusted the phrasing to stress the speculative character (lines 151, 158).

Detailed comments (referring to line numbers in the file with changes highlighted):

- Line 70: Suggest changing “contrary of” to “contrary to”

Response: Changed as suggested.

- Line 71: Suggest changing to “However, P leaching was not reduced by increased plant species diversity, ...”

Response: Changed as suggested.

- Line 94: Suggestion: “... from an experimental perspective, biodiversity and management can be manipulated simultaneously.”

Response: Changed as suggested.

- Line 114. Here the true meaning of what you are trying to say only became clear after reading the response to the comment regarding line 217 in the previous manuscript, which I can concur with. However, I would clarify the sentence by adding a few more words, for example: “However,

biodiversity and management are closely intertwined, with high nutrient concentrations sooner or later resulting in lower plant biodiversity, so that a fully crossed experiment cannot be maintained for very long in reality.”

Response: Changed as suggested.

- Line 116. “mechanistic understanding gained from experiments in observational data obtained from agricultural grasslands” is a lot to unpack. What do you mean by “mechanistic understanding”? I have the impression this is one of the most frequently misused phrases in current ecology. Furthermore, what are “experiments in observational data”? Either you do an experiment, i.e., you manipulate certain variables in order to test their effect on response variables, or you do an observational study, i.e., you study correlations among variables without manipulation. Both have their merits, but it is unclear to me what you mean here. Or do you mean something along the lines of “the combination of insights gained from both experiments and observational data”?

Response: Sorry for the confusion. The “in” (“experiments IN observational data”) referred to the “inclusion” mentioned earlier in the sentence (“inclusion of [mechanistic understanding gained from experiments] in observational data”). Nevertheless, you exactly pinpoint our intended meaning in your final statement. We rephrased the sentence accordingly (“We therefore suggest that the combination of insights gained from both biodiversity experiments and observational data of agricultural grasslands represents an alternative promising avenue”).

- Line 145. Suggest changing “increased by” to “positively related with”.

Response: Changed as suggested.

- Line 147. Suggest changing. “resulted” to “likely resulted”. This was not tested here.

Response: Changed as suggested.

- Line 149. I only find one measure in Table S2 related to organic C stocks: “Corg stocks”. How do I deduce from this table that “Corg stocks were closely related to microbial C stocks”, as the latter seems missing from the table?

Response: Because of the requested caution in relying on bivariate correlations (see response to general comment above), we deleted the reference to Table S2. Instead we cite more in-depth studies on Corg and microbial C that were conducted at the same study site.

- Line 150. Suggest changing “positively influenced” to “was positively correlated with”

Response: Changed as suggested.

- Line 151. Suggest changing “because microorganisms took up” to “as microorganisms typically take up”.

Response: Changed as suggested.

- Line 155. Suggest changing “Accordingly, a positive ...” to “Accordingly, our model indicates that a positive ...”.

Response: Changed as suggested.

- Line 161. Please indicate which result leads to the conclusion that plant C assimilation depends on plant species richness. I do not find anywhere that you measured plant C assimilation. If this is a general statement based on literature, please include references.

Response: In this part of the sentence, we wanted to point out the link between biomass production and the C cycle (via assimilation of C). The confusion probably arose from the ambiguity of what “its” did refer to in the sentence. We clarified the sentence. Now it reads: “Biomass production is linked to the C cycle and thus, the simultaneous plant species richness effects on biomass production and plant P stocks reflects the coupling of plant species richness effects on the C and the P cycle.” We consider the link between biomass production and the C cycle as textbook knowledge and did not include a reference for this statement.

- Line 166. Suggest changing “feedback on” on to “feed back to” and “Neither was there” to “There was neither a ... nor a significant relationship between P concentrations or plant biomass and AMF species richness or relative abundance”.

Response: Changed as suggested.

- Line 173. Suggest changing “In conclusion, plant ...” to “In conclusion, our data suggest that ...”

Response: Changed as suggested.

- Line 179-182. Can be collapsed into a single sentence, by deleting “We explored the mode ... experiment”.

Response: Changed as suggested (“In a first approach, we applied the SEM describing biodiversity effects on the P cycle in the biodiversity experiment to the observational data collected in the agricultural grasslands with the additional consideration of management intensity (Fig. S2).”)

- Line 184. Regarding this “direct relationship”, you propose in Table S1 that the underlying mechanism is an increased abundance of exploitative plant species. Did you not have abundance data for these particular species? If so, this could have been added as an intermediate variable in the SEM. I am just wondering, because of course this added variable would complicate the SEM so that may be a reason to opt not to include it.

Response: We agree that adding the abundance of exploitative plant species as an intermediate variable could more clearly demonstrate how this direct effect of management drives P exploitation. However, we decided not to include this variable in the model because we aimed at consistent models between the biodiversity experiment and the managed grassland. In the biodiversity experiment, filtering of the plant community towards exploitative species is reduced by maintaining the species richness levels and therefore, should not form part of the SEM.

- Line 208. Plant biomass is not in Fig. 1B, so something is missing from this explanation. Do you mean that low management intensity increased plant diversity and that this in turn lowered P stocks because plant diversity is negatively correlated with plant biomass (as indicated by Table S3)? Please clarify.

Response: We clarified the sentence. Now it reads: “In addition, low management intensity was associated with low plant P stocks (Fig. 1B) likely via a decrease in plant biomass (Table S3).”

- Line 211-214 are difficult to follow. Why is an effect on plant biomass “reversing the relationship between plant species richness and plant P stocks”? This is not at all self-evident. Basically, management changed the direction of the relationship between plant diversity and plant biomass from positive in the biodiversity experiment to negative in the agricultural grasslands (as supported by Table S2 and S3). My explanation would be that there is a unimodal relationship with biomass peaking at low diversity of a few dominant and highly competitive plant species, then dropping toward intermediate diversity and increasing again somewhat at high diversity. Then your

agricultural grasslands would be on the lower-diversity part, i.e., the negative slope, and the biodiversity experiment on the higher-diversity part, with the positive slope. Please clarify.

Response: We clarified the position of the biodiversity experiment and the agricultural grasslands in the overall relationship between plant species richness and plant P stocks/biomass: “Management-induced filtering effects in agricultural grasslands affected plant biomass and thus, reversed the relationship between plant species richness and plant P stocks as compared to the biodiversity experiment: The positive coefficient in the biodiversity experiment (Fig. 1A) covering species richness levels from monoculture to complete native communities (1-, 2-, 4-, 8-, 16- and 60-species mixtures) turned to a negative coefficient in agricultural grasslands (Fig. 1B) characterized by plant species richness which was comparable to the upper end of the experimental species richness levels (14 to 56 species per grassland) passing the filtering by varying management.” These findings do not confirm the assumption of the reviewer. Instead, experiments in which species richness and fertilization were simultaneously manipulated in a factorial way have shown that the instantaneous effects of species richness and fertilization are both positive over the whole spectrum of tested plant species richness levels from 1 to 60^{24, 25, 26}. However, in the long-run the management (particularly the fertilization) effect overrules the biodiversity effect. To still detect a possible biodiversity effect on biomass production/P exploitation in fertilized grasslands would require the selection of agricultural grassland plots with comparable N and P availability but varying species richness, which was not possible with our plots (and may be is principally impossible in agricultural grasslands).

- Line 230. But at the same time, reduced management intensity was bad for P exploitation if you look at the pathway via Corg stocks and Microbial P stocks. In such situations, calculating the combined effect may help shed light on which pathways are most important. Here, the net effect mediated through plant richness and AMF abundance is: $-0.64 \times 0.25 - 0.29 \times 0.31 = -0.25$. This is ignoring the non-significant paths for now, which actually should be taken into account as well. The net effect mediated through Corg and Microbial P stocks would then be $0.24 \times 0.54 \times 0.52 = 0.067$, which is indeed probably not significant, so I would agree with your conclusion. It may still help to mention that the latter effect is much smaller for clarity.

Response: We thank the reviewer for this suggestion. We calculated all combined effects and added the requested comparison preceding the concluding sentence of this paragraph: “The management effect on total P exploitation was mainly mediated via AMF and plant species richness (combined effect of all paths leading to P exploitation via AMF and plant species richness = -0.22), while the combined effect via Corg stocks was negligible (0.002).”

- Line 232. What I was missing after reading this paragraph is a possible explanation for the lack of significant relationship between Plant P stocks and plant P exploitation, which seems unexpected.

Response: Thank you for pointing to this omission. We included an explanation for the unexpected results (“Furthermore, the significant relationship between plant P stocks and P exploitation in the biodiversity experiment disappeared in the agricultural grasslands (Fig. 1). Instead, there was a negative relationship between plant P stocks and microbial P stocks (Fig. 1). We suspect that microbes partly outcompeted plants in terms of P uptake⁴⁶ which might have been reinforced by less cooperative AMF species that preferentially retain P for themselves^{32, 47} and thus, increased microbial P stocks at the expense of plant P stocks.”).

- Line 394. Based on Fig. S2, it seems like the paths from Corg stocks to P exploitation and from Plant richness to Plant P stocks were added afterwards and thus should also be depicted by double-lined arrows.

Response: Sorry for this lapse, Fig. 1 and the method description are correct, just the two paths (from Corg stocks to P exploitation and from Plant richness to Plant P stocks) were missing in Fig. S2. We corrected Fig. S2 in the revised version.

- Line 586. Let me copy my original remark and response of the authors: “Line 406. Why are these averages considered more “robust”? I would argue that by averaging, information is lost. Why not use both years’ data in a single model with year as an effect (random or fixed)?”

Response: The P cycle responds to changing conditions with some delay because of the equilibrium among different P pools in soil which is partly only reached with a slow kinetics (Helfenstein et al. 2018, Nature Communications 9: 9). Consequently, management effects could manifest year(s) later. Therefore, we do not regard relations among variables just within one year to be reliable.”

I fully agree that relations among variables just within one year are less reliable, that is why in my original comment I suggested to include values for both years, and include the year effect into the model, as this will preserve the variation among both years, which you lose when you average across both years. If you decide not to, it would be nice to see some argumentation as to why you think this temporal variation is not relevant in your study.

Response: We have tried to explain the challenges to account for the temporal variation in the P cycle in the previous version already. We now inserted a statement in the method section (“We considered the mean land-use intensity of the years 2011 and 2014 as more robust than the single measurements alone, because the P availability in soil is determined by the site conditions and the long-term land use, particularly in the absence of mineral P fertilization as was the case at all our study sites, except two.”). Furthermore, there were robust correlations between years for the different variables (correlation coefficient of up to $r = 0.86$, $p < 0.001$). Yet, there was a tendency towards less close relationships of variables related to soil microorganisms (e.g., $r = 0.31$, $p = 0.002$ for microbial P stocks) than of those related to plants (e.g., $r = 0.68$, $p < 0.001$ for plant P stocks). We suspect that this difference might be due to the spatial heterogeneity differing between soil and plants (because of e.g., different spatial scales of the sampling procedures). Therefore, we think that the inclusion of the years does not necessarily capture the temporal variation only and accordingly, is difficult to interpret. The averaged values then represent better the properties of the plots.

- Line 735. Statistical analyses. I much appreciate the changes made to the statistical analyses, which I think have improved significantly as a result. Still, I have a few smaller questions:

Response: We would like to thank the reviewer for acknowledging the changes made to the statistical analyses.

1) Your previous manuscript mentioned transformations of variables to improve residual normality and linearity of relationships. This is missing now from the new manuscript. I can imagine that as you used piecewise SEM, you now were able to fit generalized linear (mixed) models with data distributions better matching the actual data. If so, I think you should include a list of the individual models fitted, their data distribution (normal, Poisson, binomial etc.) and their link functions (identity, log, logit etc.). This could for example be added as a column to Table S1.

Response: We apologize for not mentioning the data transformations to meet the assumptions for the linear mixed effects models, and the resulting misunderstanding. In the revised version we now state: “Normal distribution of the variables and the homoscedasticity of the models was visually inspected and if necessary, variables were log-transformed to meet the prerequisite for statistical analyses (Tables S2, S3).”

2) I personally would have generated spatial (spline) correlograms of residual correlation versus distance among sampling points, but I see no real issue with the approach of just visually inspecting plots of model residuals versus easting and northing.

Response: Agreed.

3) It is great that you have adjusted the many Pearson correlations for multiple testing, but as I also said in my original comment, not just the linear models that make up the SEM, but also the Pearson correlations assume independence of collected data. In that sense, your p-values of the correlations, even after correcting for multiple testing, may still be artificially low because of overestimated numbers of independent samples per correlation. The only alternative would be linear models for these variables as well, similar to the ones in the SEM, and checking for spatial autocorrelation here too. I will not make this a make-or-break point for this manuscript, but just think you should be aware that you should interpret the correlation p-values in Table S2 and S3 with caution.

Response: We fully agree. Therefore, we reduced the reliance on bivariate correlations throughout the manuscript and handled the remaining ones very cautiously (see also response to the general comment).

- Line 915. Table S1: I really appreciate the addition of this Table as it provides the necessary context for and reasoning behind the hypothesized pathways of the SEM. I just noticed the paths from microbial and plant P stocks to P exploitation are not mentioned here. I realize these may be rather self-evident, but it still would not hurt to include them here, even if the explanation of these two paths is somewhat self-explanatory.

Response: Included as suggested.

- Line 920. PUE is not defined in the table caption. I would also define biomass as “plant biomass” for clarity, also in Table S3.

Response: Included (PUE definition) and changed (plant biomass) as suggested (Tables S2 and S3).

- Line 925. LUI is not defined in the table caption.

Response: Definition of LUI (land use intensity) inserted in caption of Table S3.

- Line 936. Table S5: Are these R-squared values from the final SEMs? Please clarify.

Response: The R-squared values are from the final SEMs. We clarified this in the table caption.

Reviewer comments, third round –

Reviewer #1 (Remarks to the Author):

I think the authors did a great job further improving their manuscript. At this stage I only have a few very small comments left (see below), which do not stand in the way of recommending this manuscript for publication in Nature Communications. I leave it to the authors to decide whether or not to take them into account.

Comments:

- Line 215. I do not understand what is meant by “passing the filtering by varying management”. Other than that, it is interesting to see that the positive relationship between plant species richness and plant P stocks in the diversity experiment becomes negative in the agricultural grasslands, and that this is not caused by the agricultural grasslands having lower diversity of a few highly competitive species, as I suggested. However, the explanation involving plant biomass is still not entirely clear to me. I can follow that plant biomass is related to plant P stocks, and that because management intensity increased plant biomass, it also increased plant P stocks. That would be the blue direct arrow in Fig. 1b. I can also understand that high management intensity lowers species richness, but then why is lower species richness resulting in higher P stocks? Your explanation in the rebuttal letter suggests that fertilization effects overrule diversity effects, but would such fertilization effects not already be captured by the direct arrow in Fig. 1b between management intensity and plant P stocks? The model explicitly teases out the relationship mediated by plant species richness. My point is that effects of missing variables (such as fertilization) should be captured by the direct path, unless the missing variable is highly correlated with plant species richness, and similarly responds negatively to management intensity and itself negatively relates to plant P stocks. In short, you convinced me that my hypothesized explanation is not valid, but you have not convinced me yet of any potential mechanism behind the pattern you observe here.

- Line 478. I appreciate your explanation as to why you decided to use averages across both years. You could actually use the information you provide in the rebuttal letter that variables were significantly correlated among both years to argue why you used averages. To me, the fact that they are correlated is an indication that year-to-year variation is not huge and therefore data can be collapsed into a single average over time. I leave this decision to the authors.

- Table S2 and S3. I suggested earlier to change “biomass” to “plant biomass” for clarity, and the authors say they did so, but it still seems to show as just “biomass” in my copy of the revised manuscript.

Point-by-point response

- Line 215. I do not understand what is meant by “passing the filtering by varying management”.

Response: We rearranged and rephrased the sentence for clarification.

Other than that, it is interesting to see that the positive relationship between plant species richness and plant P stocks in the diversity experiment becomes negative in the agricultural grasslands, and that this is not caused by the agricultural grasslands having lower diversity of a few highly competitive species, as I suggested. However, the explanation involving plant biomass is still not entirely clear to me. I can follow that plant biomass is related to plant P stocks, and that because management intensity increased plant biomass, it also increased plant P stocks. That would be the blue direct arrow in Fig. 1b. I can also understand that high management intensity lowers species richness, but then why is lower species richness resulting in higher P stocks? Your explanation in the rebuttal letter suggests that fertilization effects overrule diversity effects, but would such fertilization effects not already be captured by the direct arrow in Fig. 1b between management intensity and plant P stocks? The model explicitly teases out the relationship mediated by plant species richness. My point is that effects of missing variables (such as fertilization) should be captured by the direct path, unless the missing variable is highly correlated with plant species richness, and similarly responds negatively to management intensity and itself negatively relates to plant P stocks. In short, you convinced me that my hypothesized explanation is not valid, but you have not convinced me yet of any potential mechanism behind the pattern you observe here.

Response: We rearranged and extended the explanation of the negative relationship between plant species richness and plant P stocks in the agricultural grasslands (see lines 213-227 in revised version). Notably, we stick to the perspective used for the argumentation in this paragraph namely the focus on low management intensity and associated high plant species richness. We argue that there are direct relationships between management intensity and plant P stocks likely mediated via a decrease in biomass e.g., due to lower fertilization rates. In addition, low management intensity is associated with high plant species richness and thus, the coexistence of plant species with low biomass and low nutrient concentrations. Accordingly, there is also an indirect effect of management via plant species richness that is visible in the negative relationship between plant species richness and plant P stocks. This path represents the management-induced filtering effect on plant species richness.

In other words, in the experiment, increasing species richness increases plant biomass because of complementarity and facilitation, but the plant communities were randomly composited resulting in a similar likelihood for species with high and low tissue P concentrations at all species-richness levels. In managed grasslands, however, management intensity increases specifically remove those plants that are adapted to low nutrient supply and show low biomass and low tissue P concentrations. As a consequence, increasing biodiversity increasingly involves P-poor plant species with a resulting negative effect on the P stocks in plant biomass.

- Line 478. I appreciate your explanation as to why you decided to use averages across both years. You could actually use the information you provide in the rebuttal letter that variables were significantly correlated among both years to argue why you used averages. To me, the fact that they are correlated is an indication that year-to-year variation is not huge and therefore data can be collapsed into a single average over time. I leave this decision to the authors.

Response: We inserted the information on correlations between the two years (lines 330-332 in revised version).

- Table S2 and S3. I suggested earlier to change “biomass” to “plant biomass” for clarity, and the authors say they did so, but it still seems to show as just “biomass” in my copy of the revised manuscript.

Response: Sorry for this lapse, we added plant biomass in the first row but not in the first column. We now corrected this.